# Hysteresis stabilizes dynamic control of self-assembled army ant constructions

Helen F. McCreery [1✉], Georgina Gemayel [2], Ana Isabel Pais [3,4], Simon Garnier [3] & Radhika Nagpal [1,5,6]

Biological systems must adjust to changing external conditions, and their resilience depends on their control mechanisms. How is dynamic control implemented in noisy, decentralized systems? Army ants' self-assembled bridges are built on unstable features, like leaves, which frequently move. Using field experiments and simulations, we characterize the bridges' response as the gaps they span change in size, identify the control mechanism, and explore how this emerges from individuals' decisions. For a given gap size, bridges were larger after the gap increased rather than decreased. This hysteresis was best explained by an accumulator model, in which individual decisions to join or leave a bridge depend on the difference between its current and equilibrium state. This produces robust collective structures that adjust to lasting perturbations while ignoring small, momentary shifts. Our field data support separate joining and leaving cues; joining is prompted by high bridge performance and leaving by an excess of ants. This leads to stabilizing hysteresis, an important feature of many biological and engineered systems.

[1] School of Engineering and Applied Sciences, Harvard University, Boston, MA 02134, USA. [2] University of Southern California, Los Angeles, CA 90007, USA. [3] Department of Biological Sciences, New Jersey Institute of Technology, Newark, NJ 07102, USA. [4] Rutgers University, Newark, Newark, NJ 07102, USA. [5] Wyss Institute for Biologically Inspired Engineering, Boston, MA 02115, USA. [6] School of Engineering and Applied Sciences, Princeton University, Princeton, NJ 08544, USA. ✉email: hmccreery@gmail.com

At all scales, biological systems must constantly respond to changing conditions. The mechanisms controlling these systems—e.g., the central nervous system's control of breathing[1] or the behaviors regulating temperature in honey bee hives[2–4]—affect their ability to adapt in real time, and, therefore, their overall resilience. Decentralized systems face particular control challenges because noisy sensing, decision-making, and actions must all be coordinated across dispersed, sometimes loosely connected individuals or elements. Emergent self-assembly of such systems is a fundamental structuring process in biology which generates complexity and function. For example, amino acids self-assemble into DNA molecules, cells self-assemble into multicellular organisms[5–7], and some organisms self-assemble into larger, super-organismic structures, such as slime mold fruiting bodies[8], weaver ant chains[9], and fire ant rafts[10]. These structures are robust to varying conditions, for example, clusters of swarming honey bees change shape in response to changing environmental conditions, like wind[11]. This suggests the presence of control mechanisms that smooth over changes in the external environment to maintain system function. We are still discovering how such control mechanisms operate for systems across biological scales.

Nomadic army ants in the *Eciton* genus exhibit particularly impressive self-assembly. They routinely form entire nest structures out of themselves (called "bivouacs")[12–14], and they dynamically and plastically control the temperature within[14]. They also form a variety of structures to protect and ease the extremely heavy traffic flow of their foraging and migration trails[15,16]. These include bridges, ladders, ramps, scaffolds, pothole plugs, and defensive walls, which they build over rough terrain[15,17–23]. These structures are highly flexible, responding both to the amount of traffic moving over them and to the geometry of the surrounding terrain. We focus on bridges as examples of self-assembled, super-organismic structures that are particularly responsive. These bridges adjust in size as traffic flow changes[24]; ants join structures to support heavier traffic, making the bridges larger, and ants leave structures as traffic reduces, shrinking them or disassembling them altogether. This flexibility even allows bridges to recover from being forcibly broken. Rather than being based on a template, bridges are unique, forming as needed to fit the terrain, and responding to a trade-off between the number of ants making up a bridge and the path distance that it saves[25,26].

Previous work has provided insight both into how bridges adjust to changes in traffic and into the diversity of structures that are formed in different static terrain geometries[20,24–26], but the response of these structures to dynamically changing terrain geometry remains unexplored. Yet, ants form structures on surfaces which are frequently disturbed, like twigs and leaves. Such disturbances have several causes, and in some cases are small and fleeting; for example, a leaf acting as the anchor for one side of a bridge may shift back and forth by a few millimeters or less in the breeze. It may be adaptive to avoid adjusting to such momentary changes if possible. Yet bridges also face larger and more permanent disturbances, for example from animals moving through the canopy and leaf litter.

This poses a dynamic control problem; can living bridges remain functional and efficient as the gaps they span change in size? If so, how large a shift can they adjust to and how quickly? Can they respond to enduring shifts while ideally avoiding constant back and forth adjustments to smaller, momentary shifts? Bridges change in size and/or shape as ants are added or removed. However, this dynamic control must emerge solely from the distributed decisions and actions of the ants that are currently part of a bridge (potential leavers) as well as those walking on the bridge and nearby trails (potential joiners).

In this work, we combine behavioral models with manipulative experiments of living bridges to understand how these structures respond to the control problem of terrain disturbance. We first present the results of our experiments investigating how the collective structures adjust as a whole to terrain shifts. We discover that bridges are characterized by structural hysteresis as we will discuss. We then present simulation results from data-driven, mechanistic models of these collective dynamics, to explore how the actions of decentralized individuals adaptively control these structures. In particular, we focus on whether bridge construction is controlled by linear or nonlinear feedback loops. Finally, we return to our experimental data, which we used to identify proximate cues for individual decisions.

## Results and discussion

**Field experiments: collective structures.** We found that self-assembled *Eciton hamatum* bridges adaptively adjust in response to shifts in the terrain on which they are built. Detailed methods are included in Methods: Field experiments. Briefly, we moved foraging trails onto an apparatus where we could introduce a terrain gap. We repeatedly changed the size of this gap by first incrementally increasing it to 30 mm, by 1 mm every 30 s, and then incrementally contracting it at the same rate (See Fig. 1, Methods: Field experiments, and Supplementary Movie 1). As the size of the gap was expanded (the period before the dotted line in Fig. 2a, b) both the volume and number of ants increased to mean maximum values of 1080 mm³ (standard error, s.e. 84) and 18.9 ants (s.e. 1.6), respectively. Ants typically began forming a bridge when the gap was ~5 mm. As the gap size was decreased (period after dotted line), volume and the number of ants decreased back to zero as ants left the bridge. These broad dynamics across the ten complete trials were similar (Fig. 2a, b, inset panels and Supplementary Figs. 2, 3). Additionally, bridge volume (Fig. 2a) strongly correlated with the number of ants in the bridge (Fig. 2b), indicating that the density of ants per unit volume in these structures is relatively consistent (Pearson correlation coefficients range from 0.88 to 0.98 across the ten trials, see also Supplementary Fig. 4). Bridges broke and quickly reformed in eight of the ten trials; breaks occurred in both experimental phases, and these broken periods were excluded from analyses. Overall, these results show that bridges adjust dynamically to changing terrain geometry, as stretching the bridges caused them to become larger, with more ants, and contracting bridges caused them to become smaller, with fewer ants.

However, these changes were not symmetric—adjustments in the contraction phase were not the inverse of adjustments in the expansion phase. We found consistent hysteresis in several metrics; for a given gap size, bridges were larger and made up of more individuals during the contraction of the gap than the expansion (Fig. 2c, d; $t$-test for volume: mean extent of hysteresis = 0.43, 95% CI = 0.29 to 0.58, $t = 6.7$, df = 9, $p < 0.0001$; $t$-test for number of ants: mean extent of hysteresis = 0.28, 95% CI = 0.13 to 0.43, $t = 4.2$, df = 9, $p = 0.002$; see Methods: Data Analyses). This "excess" mass went into an increased mean cross-sectional area in contraction-phase bridges (Fig. 2e; $t$-test: mean extent of hysteresis = 0.40, 95% CI = 0.26 to 0.53, $t = 6.7$, df = 9, $p < 0.0001$) and caused contraction-phase bridges to hang lower, having more slack, than expansion bridges (Fig. 2f; $t$-test: mean extent of hysteresis = $-0.27$, 95% CI = $-0.34$ to $-0.21$, $t = -9.1$, df = 9, $p < 0.0001$). Bridges had the most slack midway during the contraction (Fig. 2f), suggesting that as the gap begins closing the bridge does not immediately shorten, instead of hanging lower until shortening later in the contraction phase. Hysteresis in each of these metrics was present in every trial, regardless of variance in traffic or the presence and timing of breaks.

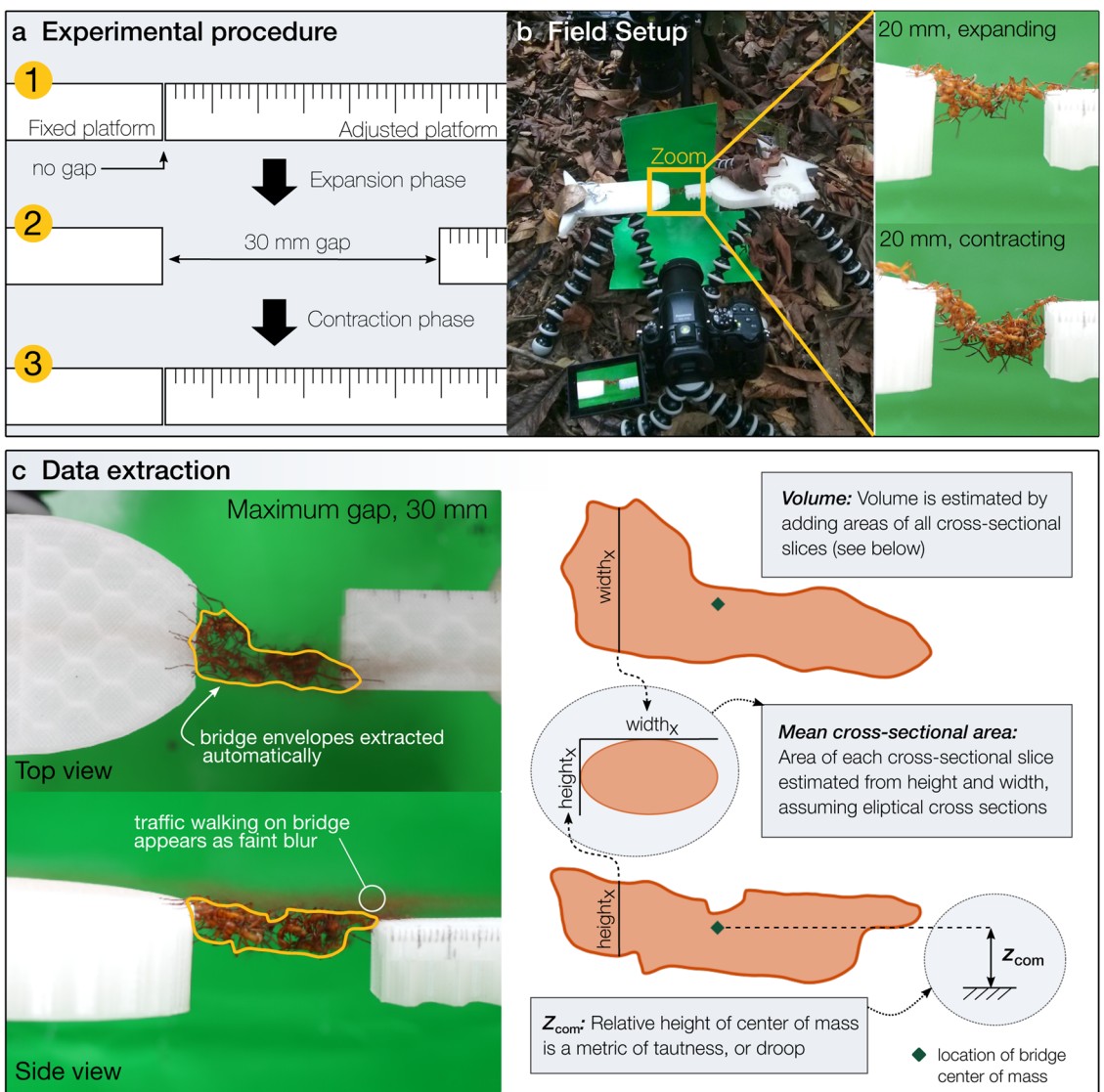

**Fig. 1 Experimental procedure and data extraction summary.** Experiments were conducted on robust *E. hamatum* foraging trails, which were moved onto the experimental apparatus while it was closed. **a** Experimental procedure: The size of the gap was increased by 1 mm every 30 s until the gap reached 30 mm (expansion phase), then decreased at the same rate till no gap remained (the contraction phase). **b** Field setup: Experiments were recorded from both the side and the top, examples of bridges during each phase of the same trial are shown. **c** Data extraction: Example images and silhouettes from the maximum size bridge (30 mm) of the same trial as the images of 20 mm bridges shown in panel a. The envelopes of the bridges were extracted at a temporal resolution of 1 s; for each focal second, image frames were averaged over 10 s to remove ants walking on the bridge from the extracted envelopes. Envelopes were automatically extracted using hue-saturation-value (HSV) thresholding, with thresholds checked independently for each trial due to lighting differences. Locations of fixed points on the platform were used to re-scale and combine data from the side and top views into a single coordinate system in which 100 pixels = 1 cm. Estimates of bridge volume, mean cross-sectional area, and relative height of the center of mass were recorded from the extracted envelopes as shown. See Methods: Data extraction and Supplementary Note 1 for additional details of the data extraction process, including additional bridge metrics.

This hysteresis cannot be explained by traffic, which is known to affect bridge structures[24], as traffic was independent of phase (Supplementary Fig. 9). One may speculate that hysteresis emerged because we changed the gap size too rapidly for the bridge to keep up. Yet this is also an insufficient explanation, as earlier work demonstrated that ants in bridges adjust to changes in traffic flow within seconds[24]. We also excluded the possibility that delayed response to geometric changes causes hysteresis, as discussed in our mechanistic models section, below. Thus, the state of these structures depends not only on gap size and traffic, which was already known but also on the history of the structure, which has strong implications for our understanding of the control mechanism. Additional measures of the bridges, as well as

full-size plots for each trial, are included in the supplementary information (Supplementary Figs. 2–8).

Hysteresis is a type of structural memory[27], because the current state (i.e., number of ants) depends not only on the value of a driver (i.e., gap size) but also on the history of the system. Hysteresis is an important feature of many biological systems, including cellular and genetic systems[28,29], neurological systems[28,30], animal groups[31–34], and ecosystems[35,36]. While hysteresis in some contexts is a signal of reduced resilience (e.g., ecosystems[37]), it is a key stabilizing feature of other systems, especially systems designed or evolved for control. Hysteresis can allow for multiple stable states (bistability)[33,38,39] and memory[39,40] and can reduce the impact of noise[41]. Biological

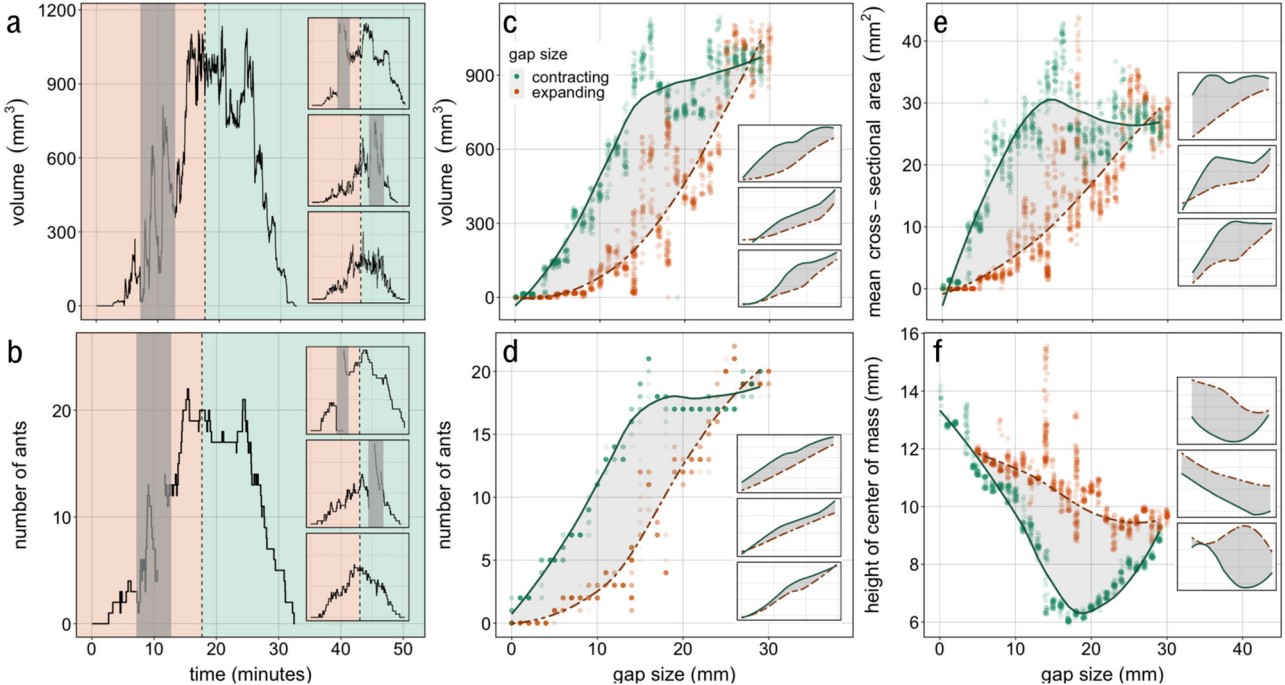

**Fig. 2 Changes in collective structures in experiments. a, b** Volume and group size of self-assembled bridges: **a** Estimated volume of collective bridge structures over time for one focal trial (main figure) and three other examples (inset). The dotted vertical line indicates the time when the experiment shifted from the expansion phase (increasing gap size) to the contraction phase (decreasing gap size). Gray shading indicates that the bridge was broken or recovering from a break; result metrics may be inaccurate during these periods and they were, therefore, excluded from analyses. **b** The number of ants in the bridge structure over time for the same focal trial (main figure) and three other examples (inset). **c–f** Hysteresis: Trials consistently show hysteresis, with bridge status at a particular gap size differing during the expansion and contraction phases, for volume (**c**), number of ants (**d**), mean cross-sectional area (**e**), and tautness, or the height of the center of mass of the bridge from the side view (**f**; lower values indicate bridge is hanging lower). **c–f** Panels show result metrics over gap size for the same focal trial as in panels **a** and **b**, as well as for three other examples (inset). Points show individual measurements, taken every second, lines are smoothed LOESS (local regression) for the expansion (orange points, dashed orange line) and contraction (green points, solid green line) phases. The area between the smoothed lines (shaded gray) shows the extent of hysteresis. **c, e, f**) Points are jittered to improve clarity. **a–f** See Supplementary Figs. 2, 3, 5–8 for all complete trials.

examples include cellular signaling—where both bistability and memory are associated with hysteresis[38–40,42]—and human vision —low corneal hysteresis seems to exacerbate glaucoma[43]. Hysteresis is also intentionally added to stabilize many engineered control systems, for example, to make these systems immune to noise, preventing constant on-off switching near thresholds (e.g., thermostats, Schmitt triggers[44]). In army ant bridges, hysteresis may play a similar role, improving stability and allowing the structures to avoid unnecessary adjustments to small, momentary gap size changes. We investigated this possibility as described in the following section.

**Mechanistic models**. Living army ant bridges adaptively respond to changes in the geometry of the surrounding terrain. To understand how this response is dynamically controlled, we developed three behavioral models of individual decisions to join or leave structures—two linear with respect to the gap perturbation and one nonlinear—as mechanistic hypotheses. Note that joining and leaving events do not appear to be directly prompted by changes in gap size, as the density of events is uniform across the 30-s intervals between adjustments (Supplementary Fig. 10). We focused first on whether simulations of these models under the conditions of our field experiments can reproduce the hysteresis we observed, which cannot be simply explained by factors previously reported to affect these structures, such as traffic and gap size. We then simulated these models' behavior when built on terrain that shifts slightly back and forth, as we expect these vibration-like movements to be common in the ants' environment

and hysteresis is a known mechanism for damping oscillations. We were particularly interested in whether nonlinear mechanisms are required to explain the collective bridge dynamics because nonlinear responses are an important feature of many types of complex emergent phenomena[34,45,46] and implicated in engineered hysteresis[47–50]. All three models were parameterized using the data; none of the models include any free parameters that we tuned.

*Linear models: baseline and delay*. We explored two data-driven linear models of ants joining and leaving bridges (Fig. 3). These are fixed-probability models because they rely on the assumption that individuals' probability of joining and leaving is constant during each phase. It is reasonable to expect such constant behavior, as the rate of perturbation in our experiments was constant at 2 mm/min. We simulated the number of ants in a bridge resulting from stochastic joining and leaving events, with the restriction that the number of ants leaving cannot be larger than the current number in the bridge. Each simulation timestep corresponds to a 30-s gap interval in the real experiments. The numbers of joining and leaving events in each timestep were drawn from negative binomial distributions fitted to the data (see Fig. 3a and Methods: Models). During the expansion, the mean number of joining and leaving events in each 30 s timestep were 1.13 and 0.67 (dispersion parameter, $\theta = 1.33$ and 1.98), respectively, and during the contraction, there were an average of 0.33 and 1.04 joining and leaving events ($\theta = 0.65$ and 8.10) in each step, respectively.

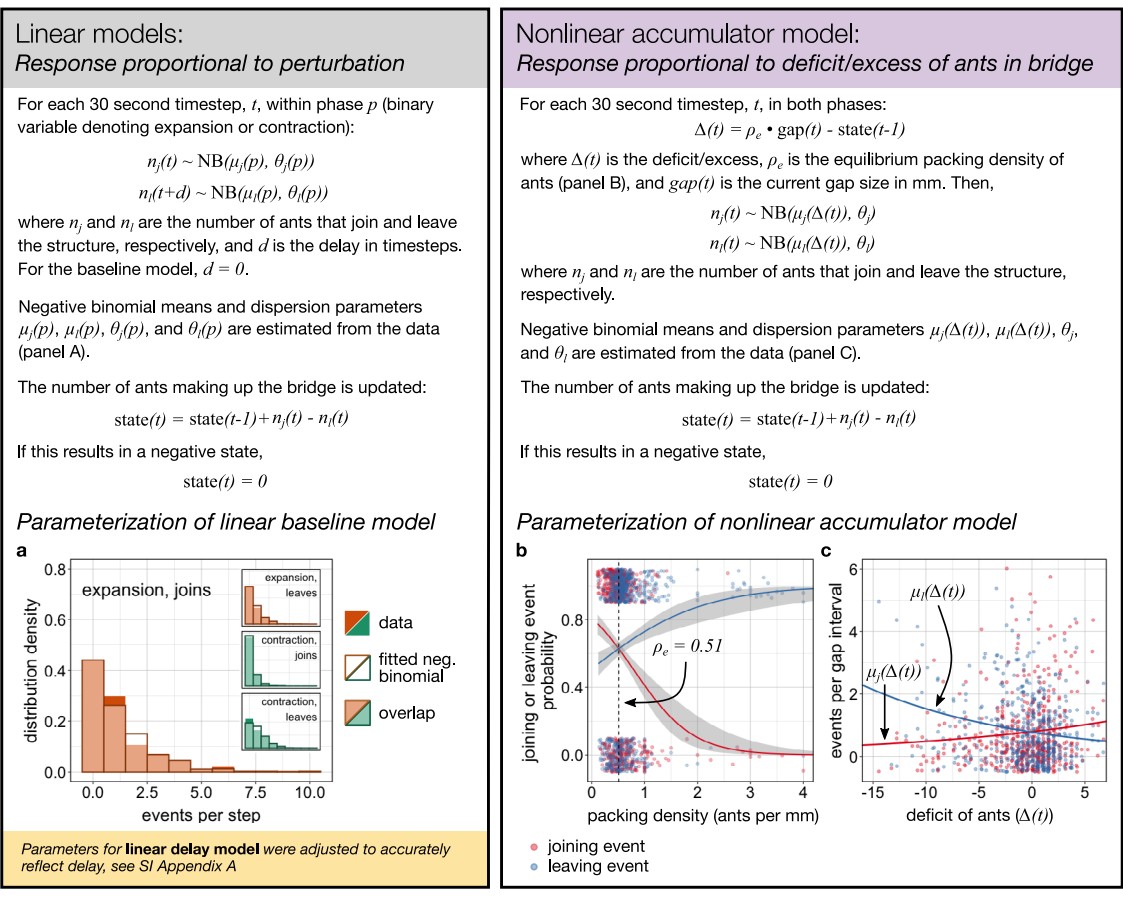

**Fig. 3 Mechanistic model descriptions.** Model descriptions with equations are shown at the top. **a** Distributions of the number of ants joining or leaving self-assembled bridge structures per 30 s gap interval, measured separately during gap-size expansion and contraction phases, shown respectively in orange and green, and negative binomial distributions fitted to these data. Main panel: distribution of joining events in the expansion phase. Inset panels show expansion leaving events and all events in the contraction phase. **b** Probability of joining (pink) and leaving (blue) at different packing densities (number of ants/gap size), determined with logistic regression. Gray bands show 95% confidence intervals. The equilibrium packing density, where the joining probability is equal to the leaving probability, is 0.51 (dashed vertical line). Dots show joining (pink) or leaving (blue) events (at 1), or gap intervals with no such events (at 0). **c** Number of joining and leaving events in each gap interval across deficit values, where the deficit is the equilibrium number of ants minus the actual number of ants in the bridge. Thick lines show negative binomial regressions fitted to these data.

Our linear baseline model assumes that ants join and leave bridges quickly after deciding to do so, while in our linear delay model there is a lag of constant duration between an ant deciding to leave the bridge and actually doing so. As discussed above, long delays may be unlikely based on known response times to traffic changes[24], yet we include this possibility because one may intuitively expect it to result in hysteresis. We modeled leaving delays of 30 s up to 3 min (1 to 6 timesteps), and we do not allow for a delay in joining, as ants that could potentially join the bridge (those walking over it), do not remain nearby for long enough to result in a joining delay of more than a couple of seconds. If there were indeed a leaving delay in real ants, then the leaving events recorded at the beginning of the contraction phase must reflect decisions to leave made during the last part of the expansion. Thus, we adjusted the distribution for leaving events in our delay model to still accurately reflect leaving decisions for the assumed length of delay (see Methods: Models). We evaluated both of these linear models by first plotting the average simulated expansion and contraction dynamics to compare with the experiments, and then by measuring the extent of hysteresis. The extent of hysteresis can range from −1 to 1; we define it as the area between expansion and contraction curves as in Fig. 2d, scaled by the area under the higher curve, and it is negative if the expansion curve is higher.

The baseline model failed to reproduce the dynamics we observed in real ant bridges. While the models' behavior was highly variable, the majority of simulations resulted in negative hysteresis, that is hysteresis in the opposite direction as the data, with more ants in the simulated bridges during the expansion than the contraction (Fig. 4a). The mean extent of hysteresis was −0.22, and it was negative in 65% of the 10,000 simulations, compared with an average of 0.28 in the real experiments, with 100% of trials having positive hysteresis (distribution of extents of hysteresis shown in Supplementary Fig. 11). The negative hysteresis produced by the linear models is due to a slight asymmetry in the data on joining and leaving in each phase. Briefly, in our experiments, the net loss of ants (number leaving the bridge minus number joining) in the contraction (specifically, 0.71) tended to be slightly higher than the net gain during the expansion (0.46, Supplementary Fig. 12). The delay model performed only slightly better than the baseline model, resulting in an average extent of hysteresis of −0.15, and the hysteresis was negative in 59% of simulations (Fig. 4b and Supplementary Fig. 11). The delay model performed poorly regardless of the length of delay, and the results of models with delays of one step up to six steps were all similar (Supplementary Fig. 13).

We further tested these models by subjecting them to small, momentary changes in gap size. We expect that these vibration-like

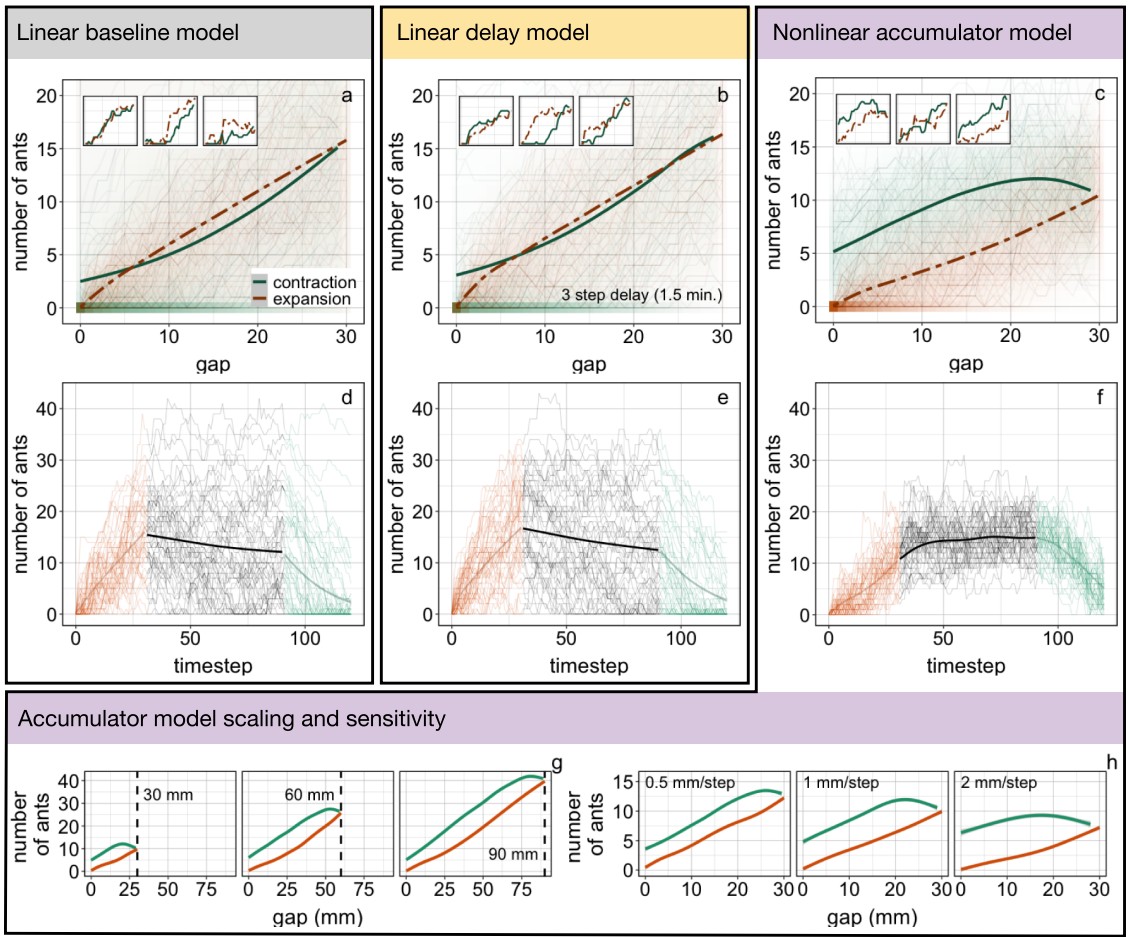

**Fig. 4 Model comparison.** The nonlinear accumulator model reproduces hysteretic dynamics of real bridges much more accurately than either linear model, avoids constant adjustments to momentary shifts, and performs well across a range of scales and rates of gap size changes. **a–c** Results of 10,000 simulations for the baseline model, delay model with three-step delay (1.5 min), and accumulator model, respectively. Thick lines indicate means across the simulations (expansion: dashed orange line, contraction: green solid line). Shading indicates the number of simulations occupying a cell (combination of gap size and the number of ants in either expansion or contraction in orange and green, respectively). Thin lines show 50 example simulations. Inset panels also show example simulations. The nonlinear accumulator model reproduces dynamics of real bridges much more accurately than either linear model. **d–f** Results of 1000 simulations for each model under the "vibration" conditions. Thick lines show means across simulations, thin lines show 50 example simulations. **g, h** Results of accumulator model scalability and sensitivity analyses, respectively. Lines show means across 100 simulations for each panel in which either the maximum gap size (**g**) or rate of gap size change (**h**) varied from the original experiment as shown.

conditions occur frequently in nature, as bridge anchors such as leaves shift slightly in the breeze. We performed 1000 simulations of experiments with three phases of movement: expansion as before, a vibration of 1 mm (gap shifts back and forth from 29 and 30 mm), and contraction as before. The behavior of both models is highly variable, with many of the simulated bridges rapidly growing or shrinking to unreasonable numbers of ants during vibration. Even though the gap size was always 29 or 30 mm throughout the vibration phase, the baseline and delay models resulted in "extreme" bridges, at some point having fewer than five ants or more than 25 ants, in 84 and 85% of simulations, respectively. Both models also tend to slowly lose ants on average during vibration (thick black line in Fig. 4d, e), which would lead to bridge failure over time; this is again due to the slight asymmetry in joining/leaving between phases, discussed above (Supplementary Fig. 12). Furthermore, in 41 (baseline) and 39% (delay) of simulations the structures reached zero ants, indicating that with the linear models, even a slight asymmetry leads to frequent failure during vibration. These models clearly do not produce structures that could reliably support traffic when facing the small, repeated shifts in gap size which are likely common in nature.

Given the uniform rate of perturbation in the field experiments, it was reasonable to expect ants to have fixed probabilities of joining or leaving in each phase. However, both linear models failed to reproduce hysteresis resembling our experimental results. These models propagate errors, in the sense that if too many ants leave in one timestep, leaving events are still equally likely in the next. Modeling a delay in leaving does not resolve this; every gap shift gets a response whether or not it should, even if that response is delayed. The linear models are also highly sensitive to asymmetry—even the slight asymmetry in the net gain/loss of ants between phases in our experiments led to these models typically producing negative hysteresis and bridges which shrank during vibrations. Finally, both linear models resulted in such variable behavior—even with small, momentary gap shifts— that any given simulation was unlikely to produce a viable structure.

*Nonlinear accumulator model.* After excluding the linear models, we chose to explore a nonlinear "accumulator" model, with parameters still drawn from the data. This is a dynamic-probability model—the chances of joining and leaving are based

on the excess or deficit in the number of ants in the bridge, that is, the deviation of the actual number from the equilibrium number for that gap size (calculated using equilibrium "packing density", or ants per mm of gap, as described below, also see Methods: Models). When the number of ants is small for the gap size (i.e., the bridge is stretched and packing density of ants is low—large deficit), ants are very likely to join a bridge and unlikely to leave. Conversely, when there are more ants in the bridge than necessary (i.e., the bridge is slack with excess ants), ants are likely to leave and unlikely to join. We refer to this as an "accumulator" model because as changes in the terrain geometry *accumulate*, the likelihood of response increases nonlinearly as described below, and small changes are relatively unlikely to produce a response. While there are endless possible nonlinear models, we chose this model to explore because it agrees with our intuition that bridge adjustments respond to the magnitude of perturbations. It also fits earlier work showing that the packing density of bridges affects individual behavior[24], and conceptually related models explain other social insect behaviors, such as response threshold[51], error correction[23], and integrator[52] models.

We parameterized our accumulator model using our experimental data. The equilibrium packing density of bridges in our experiments was 0.51 ants per mm of gap size; this is the packing density at which ants are equally likely to join as leave a bridge (Fig. 3b). Thus, a bridge spanning a 20 mm gap should be composed of about ten ants, and if our simulations result in a 20 mm bridge with only seven ants, we model a deficit of 3. We then mapped simulated differences from this equilibrium to the resulting number of joining and leaving events using our experimental data. We used negative binomial regressions of the data to predict the mean number of each type of event, µ, at each deficit value—thus, µ depends nonlinearly on the deficit/excess ($\Delta(t)$), as shown in Fig. 3c. Specifically, joining events increase with increasing deficit ($\beta = 0.049$, $\theta = 0.773$), and leaving events decrease ($\beta = -0.067$, $\theta = 1.96$); see Methods: Models for analysis details. These data also confirm that high deficits lead to more ants joining and fewer ants leaving while the inverse is true for bridges with excess ants, as expected for the accumulator model. We also note the data include more instances of extreme excess than instances of extreme deficit—while it was somewhat rare in our experiments for a bridge to have a deficit of more than five ants, it was not uncommon to have an excess of ten or more ants. This is likely because if the deficit grows too large a bridge will fail. On the other hand, while a very large excess (a highly negative deficit) may be inefficient, it is unlikely to lead to bridge failure.

The accumulator model behaves similarly to the experiments with real ants (Fig. 4c). The average extent of hysteresis was 0.48 (see Supplementary Fig. 11 for distribution of hysteresis extents). While this is higher than the extent of hysteresis in the experiments with real ants, which was 0.28, it is much closer than the linear baseline and delay models. Furthermore, 96% of the accumulator model simulations resulted in positive hysteresis, compared with 100% in the real experiments. The accumulator model also performed well in the vibration condition (gap size-shifting back and forth between 29 and 30 mm). Unlike the linear models, this model maintained a consistent number of ants on average during vibrations (thick black line in Fig. 4f), and with far less variation—75% of simulations avoided "extreme" bridges, maintaining between 5 and 25 ants for the entire vibration phase, and only 1% of simulations reached 0 ants at any point. This is because the 1 mm shifts in the vibration phase hardly change the packing density, so joining and leaving events are relatively unlikely; this is quite different from the linear models which are equally likely to respond to every 1 mm shift. Furthermore, in contrast to the linear models, in which the chance of joining and leaving events is unaffected by past behavior, the accumulator

model is self-correcting—if too many ants leave in one timestep the deficit increases, making future leaving events less likely and joining events more likely. Thus, this model's behavior inherently depends on its history, and this hysteresis damps potential oscillations near the equilibrium, making the bridges more stable.

Finally, we explored how sensitive the accumulator model's behavior is to the maximum size of the gap and to the rate of gap size change. We found that this model is scalable, as it behaves similarly whether the bridge spans a gap reaching a maximum of 3, 6, or 9 cm (Fig. 4g). The behavior of the model is somewhat sensitive to the rate of gap size change. Specifically, the extent of hysteresis is rate-dependent: when gap changes accumulate more quickly there is predicted to be more relative hysteresis (Fig. 4h), and we also predict slightly smaller bridges in this case. Yet the overall system dynamics are similar across rates.

The dynamic-probability accumulator model outperformed the fixed-probability linear models in every analysis. It consistently reproduced hysteresis in the same direction as our experiments, performing well with the relatively large, directional changes we exerted in our real experiments. It maintained stable bridges when facing small, momentary changes in gap size, avoiding constant adjustments, because it is self-correcting, which also makes it much less sensitive to asymmetry. Furthermore, it is scalable, behaving consistently across different maximum gap sizes, and not highly sensitive to the rate of gap size change. Whereas the poorly-performing linear models depend only on external dynamics (i.e., expansion vs contraction), the accumulator model incorporates nonlinear feedback that produces stabilizing hysteresis. Nonlinear dynamics also play a key role in many other mechanisms of emergent action and control, due to quorums, thresholds, critical points, and more[34,45,46], and the accumulator model is similar to, though mechanistically distinct from, an integrator model previously applied to ant foraging[52]. We conclude that this nonlinearity is an important feature of the dynamic control mechanism of collective army ant bridges, which are able to withstand small perturbations in gap size without adjusting, becoming more likely to adjust as perturbations accumulate.

**Field experiments: individual cues**. According to our model, the deviation from equilibrium packing density of ants in a bridge (i.e., magnitude of deficit or excess) can explain the number of ants joining and leaving these structures, but we do not expect ants to respond to this directly, as it may be difficult for ants to perceive. Here, we consider immediate cues causing individual ants to respond to such a deficit—how can an ant locally detect if a bridge is overstretched or slack? We investigate performance (i.e., how easy the bridge is to cross) as a likely candidate cue for ants sensing the deviation from equilibrium. We hypothesize that bridges with a large deficit (too few ants in the bridge for the gap size) would be harder to cross and have low performance. We further assume ants joining a bridge that has a large deficit will improve its performance (i.e., whether and how easily ants can walk across it), and ants leaving a slack bridge, with excess ants, will likely also make it easier and/or faster to cross. Thus, while there are many possible cues that may lead to individual joining and leaving decisions, we first consider performance-based cues. Intuitively, both joiners and leavers may have information about bridge performance; potential joiners (those walking on the trail) can likely sense how easy the bridge is to cross while potential leavers (those within the bridge) may be able to sense how many and how rapidly ants are able to cross over them[24].

Our measure of bridge performance is the flow rate of ant traffic over the bridge itself relative to what would be expected based on the current flow rate on the trail next to the bridge

(while the total flow of ants must be consistent across the trail, local flow rate changes with fluctuating traffic). We used optical flow as a proxy for the local flow rate, which integrates information about the speed and number of crossing ants, and we account for overall changes in traffic by recording performance as the signed residuals from regressions of optical flow on bridges vs optical flow on both of the adjacent platforms (Fig. 5, Supplementary Note 2, and Supplementary Fig. 14). Our measure of performance reflects the ease with which ants can cross a bridge. If a bridge slows down the flow of ants to a greater extent than usual, the performance measure will be low; high performance reflects bridges which do not slow down traffic or do so less than usual. In general, performance was higher in the expansion phase than the contraction phase (reflected in the distributions in Fig. 6). This suggests that the taut bridges occurring during expansions are easier to cross than more slack bridges in the contraction phase. We investigated whether performance is a cue for joining or leaving events by comparing performance immediately before events with performance at random intervals, using a resampling approach (see Methods: Data analyses).

*Joining mechanisms.* We found that bridge performance strongly affects ants' decisions to join, yet the direction of this effect is contrary to our expectations. We predicted low performance to occur at high deficits (fewer ants than the equilibrium number) and then lead to joining, as ants struggling to cross a bridge will slow down, and will likely then be walked on by those behind them. However, performance immediately preceding joining events in our experiments was substantially and significantly higher than would be expected by chance (Fig. 6, left). This result cannot be attributed to overall traffic patterns, which were previously known to affect individual decisions[24,25], because our performance measure accounts for overall traffic. The effect of high performance on joining is distinct from—and in our experiments, clearer than—the overall effect of traffic (Supplementary Fig. 15).

The evidence that joining events are preceded by high bridge performance initially seems to contradict our data- and model-based results that bridges with high deficits have higher joining rates. This discrepancy arises from our faulty assumption that bridges with high deficits (i.e., too few ants) will have lower performance. In fact, high deficit was not associated with low performance in our experiments (Supplementary Fig. 16), and we saw higher performance on average during the expansion than the contraction phase, suggesting that having excess ants in a bridge may be worse than having too few, at least within the range of packing densities we observed. While we expected the equilibrium packing density to be close to optimal, it seems to result in more ants than necessary for high bridge performance. Additionally, earlier work indicates that after natural bridges are broken, traffic flow reaches the previous level before the number of ants in the bridge recovers to the previous number[24]. Ants continue to join these bridges that are back to "normal" traffic flow, further supporting that ants will join bridges with high performance. Our results also strongly support that ants join bridges with high performance. This positive feedback may reflect colonies' investment in portions of the trail that are functioning well, regardless of the bridge's deviation from equilibrium. After all, ants should be capable of abandoning path sections that are not functioning to find new paths. These "extra" ants may also strengthen the bridges to minimize breaks, may provide extra capacity for sudden traffic increases[24], and may provide some insurance against the gap increasing.

*Leaving mechanisms.* We do not observe strong effects of performance on decisions to leave bridges. While performance

## a  Flow rate of ants (optic flow) extracted separately for bridge region and platform regions

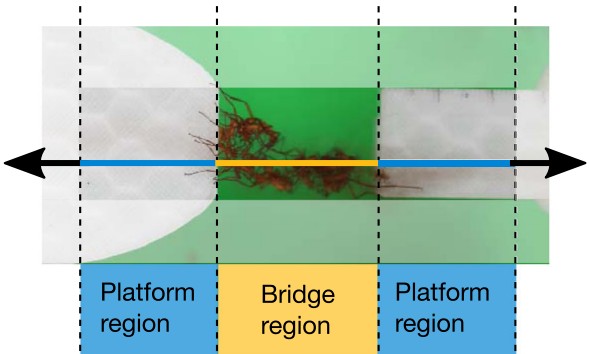

## b  Performance measure is relative flow rate, i.e. the signed residuals from regression of bridge flow on platform flow

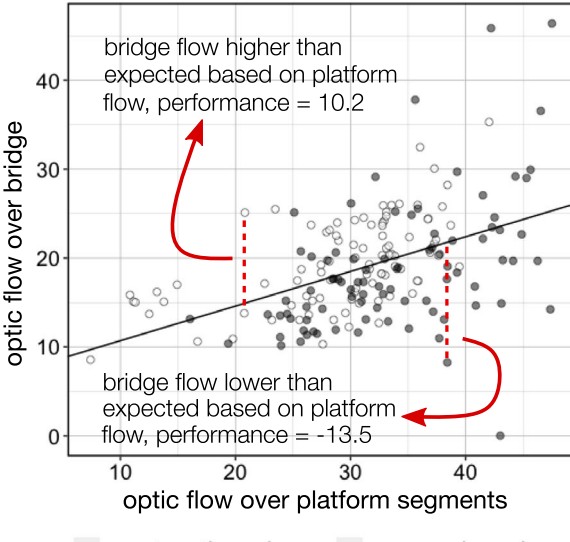

**Fig. 5 Measuring bridge performance. a** We used optical flow to measure the flow rate of ants across the platforms and bridge. **b** Our measure of performance was the flow rate of ants over the bridge relative to what would be expected based on the flow rate over the nearby platforms, i.e., the signed residuals from separate regressions for each trial of bridge flow on platform flow. Optical flow measures the amount of movement from one video frame to the next, in units of pixels per frame. Additional details of optical flow data extraction and the performance metric are included in Supplementary Note 2.

immediately prior to leaving events is significantly lower than expected during the expansion phase (Fig. 6, right), the effect is much weaker than for joining events and is sensitive to the time interval over which performance is measured (Supplementary Fig. 14). There is no such effect of performance on leaving events in the contraction phase. Thus, potential leavers do not seem to use performance as a cue.

Instead, we hypothesize that potential leavers respond more directly to a bridge's deviation from the equilibrium number of ants, by sensing a closely-linked structural feature such as how taut or slack the bridge is. Intuitively, we expect ants in a bridge to be more stretched and experience higher tension at high deficits (i.e., low packing density) than those in a bridge with excess ants.

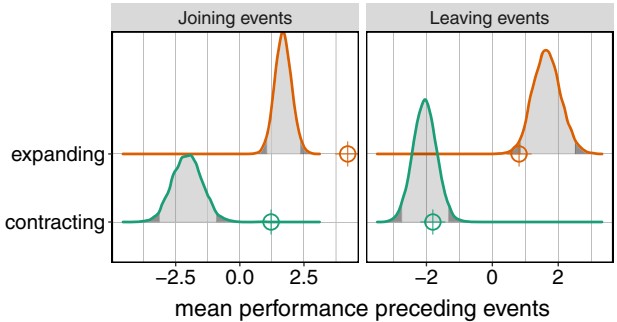

**Fig. 6 Joining events are preceded by periods of high performance.** The effects of performance (ant traffic flow over bridge relative to expected flow based on traffic over platforms) on joining and leaving events were analyzed with a resampling approach. Distributions (density plots) show the mean performance during the 10 s preceding events at randomized times, from 10,000 simulations, for joining events (left panel) and leaving events (right panel) during the expansion (orange) and contraction (green) phases (resampling analysis described in detail in Methods: Data Analysis). Dark gray shading in the tails of each distribution indicates the most extreme 5% of that distribution (below 2.5% and above 97.5%). The corresponding actual means from the experiments are shown as points (open circles with crosshairs). Performance preceding joining events is significantly and substantially higher than expected by chance. Performance preceding leaving events is marginally lower than expected during the expansion, and not different than expected during the contraction. Our conclusions for joining events are not sensitive to the time interval over which performance is measured (Supplementary Fig. 14 and Supplementary Note 2).

We explored this hypothesis at the level of the entire structure and found that the tautness (height of the center of mass) of an entire bridge does not predict leaving events in either the expansion or contraction phases (Supplementary Fig. 17).

However, this global analysis is insufficient, because structural or mechanical cues experienced by an ant will vary depending on where the ant is located within a bridge and the network of grasping connections in the structure (which is known to affect the probability of leaving[24]). We, therefore, expect considerable variation in the tension (or other mechanical factors) experienced by individuals within a structure; for example, even in a slack bridge ants at the edge acting as anchors may experience high tension. We leave a direct test of the hypothesis that structural cues prompt leaving events to future studies that allow for careful modeling of the forces on individual ants. Nevertheless, this hypothesis has some indirect support. Consider that in addition to ants experiencing lower tension on average when a bridge is slack, there should also be a reduction in tension when the bridge is supporting less mass, i.e., when traffic on the bridge is lower. We know from earlier work that ants are more likely to leave when they are walked on less[24], and this is further supported by our experiments (leaving events associated with low absolute bridge traffic, Supplementary Fig. 18). If potential leavers use reduced tension as a cue it may account for both the known effect of traffic on leaving and our results that ants are more likely to leave when there are excess ants in the bridge.

This points to a clearer explanation of our results, with two separate mechanisms accounting for joining and leaving (Fig. 7). As discussed above, we know that *E. hamatum* workers join bridges with high relative traffic flow (high performance), suggesting that colonies may invest in high-functioning sections of trail. This does not contradict our model, as plenty such high-performance periods will occur when the deficit is also high; large deficits do not lead to low performance as we assumed. As ants

join a bridge the deficit is reduced, and if excess ants join it may prompt those ants or others to leave, perhaps because they experience low stretch or tension due to the extra ants. Joining and leaving mechanisms do not *both* need to be calibrated such that bridges with too many ants get smaller and those with too few get larger, if the combined effect of joining and leaving produces this response. Thus, by potential joiners and leavers responding to different cues, these dynamic bridges can both adjust in size to reflect traffic requirements and respond to trail dynamics at a larger scale, allowing investment in path options that perform well. Army ants also use distinct sets of cues for joining and leaving defensive walls that form to separate trails from competitors[22].

While these results explain how well-functioning bridges respond to shifts in terrain geometry, it is less clear from our study how poorly-performing bridges recover from periods of high deficit. Such periods could be rare if trail routes shift so gradually that performance stays high; yet we still intuitively expect them to occur, for example when a bridge is first needed. What causes ants to join bridges with too few ants and low performance? One potential explanation is that there are one or more other joining cues in addition to high performance, such as social cues or the presence of pot-hole-like gaps in the bridge surface[18]. By focusing more on bridges as they initially form, including the recovery of bridges from breaks, future experiments could determine whether additional joining cues are important.

*Eciton hamatum* army ants self-assemble into remarkably flexible, living bridges, capable of adjusting to constant changes in real time, as structures rapidly adapt to terrain disturbances, forming and growing to 3 cm in just 15 min in our experiments. These adjustments allow colonies to maintain robust trails with heavy traffic in unpredictable conditions, and the response mechanisms may extend to other army ant structures, including ladders, ramps, scaffolds, pot-hole plugs, and bivouacs[15–20,23]. At a global level, army ant colonies can behave like living organic materials, exhibiting cooperative behaviors that are analogous to those seen across scales, including in morphogenesis, wound healing, and the assembly of slime mold fruiting bodies[8,53,54]. For example, the bridges we studied add and remove material (ants) as needed to maintain the high flow of ants on the trail; in this sense, their assembly resembles epithelial tissue morphogenesis, in which cell division is induced in stretched tissue (adding material), and apoptosis is induced in slack tissue (removing material)[5,6].

We examined how living army ant structures respond to the dynamic control problem of the ground moving beneath them, focusing on whether and how the structures adapt, what the underlying control mechanism is, and how this control is implemented. In nature, these bridges face both long-lasting terrain shifts from sources such as animals moving through the leaf litter as well as frequent small, vibration-like movements as anchors move back and forth in the breeze. We found that the number of ants making up a bridge is controlled collectively by ants moving along the trail—who join high-performing bridges— and by ants making up the bridge—who leave when a bridge has too many ants. These separate mechanisms for joiners and leavers balance one another and lead to hysteresis, an important feature in the control of both biological and engineered systems, which adaptively stabilizes the bridges, preventing constant switching between ants joining and leaving. Our accumulator model reproduces overall patterns similar to real bridges including this hysteresis—which our experiments showed is a defining feature of the system—and is self-correcting; producing robust bridges which stably adjust to lasting terrain shifts while disregarding momentary, vibration-like movements. By illuminating the presence and benefit of stabilizing hysteresis in army ant bridges,

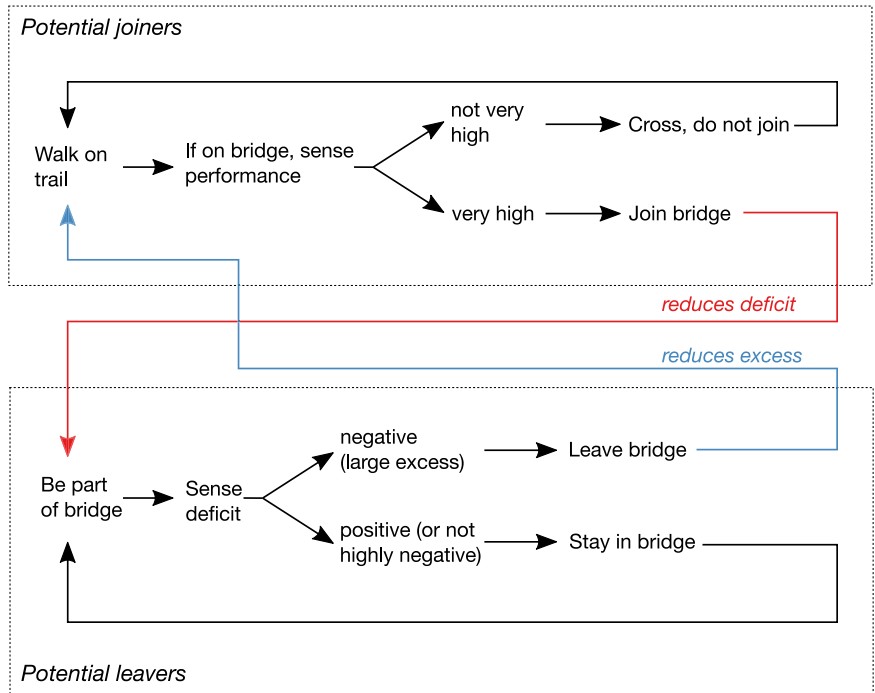

**Fig. 7 Flow diagram for individual ants' decisions.** The evidence supports separate mechanisms leading to ants choosing to join and leave bridges. High bridge performance increases the likelihood of ants joining the bridge, and the likelihood of ants leaving seems to increase when there are too many ants (large excess).

our work demonstrates the potential of related hysteretic mechanisms for stable control of decentralized systems—in which noisy sensing, decision-making, and actions must all be coordinated across dispersed individuals. Fundamentally, such decentralized control drives the success of these systems across scales, from subcellular organization to superorganisms.

## Methods

**Field experiments**. Experiments were conducted on robust *E. hamatum* foraging trails on Barro Colorado Island (BCI), under project number 3920, approved by the Smithsonian Tropical Research Institute on February 16, 2016. Because bridges tend to disassemble when traffic on the foraging trail drops, we chose foraging trails with substantial traffic that was relatively consistent. If we were able to locate multiple trails we set up our experiments on the most robust one. Our experimental apparatus consisted of two 3D-printed platforms mounted on small, flexible tripods (JOBY GorillaPods). One static platform consisted simply of a flat surface, while the other platform was adjustable; a simple gear mechanism allowed us to shorten or lengthen a beam to open or close a gap between the platforms by manually turning a wheel. We set up the platforms over the top of the foraging trail so that the walking surfaces were flat and with the beam extended so there was no gap between the platforms (see Fig. 1).

We then moved the ants' foraging trail onto our apparatus by constructing ramps up onto and down from the platforms out of leaves and twigs that ants were already walking on (which had trail pheromone on them). We also completely removed the part of the foraging trail underneath the platforms by disturbing the leaf litter, to prevent ants from bypassing our apparatus. We physically moved ants from underneath the apparatus to the tops of the platforms, while other ants reached the platforms using the ramps we constructed. Ants on top of the platforms gradually found their way across and down our ramps, building up trail pheromone on top of the platforms. After sufficient time, the trail over the platforms was robust, with ants moving from the trail in the leaf litter up one of the platforms, crossing to the other platform, and moving back down the other ramp.

After the ants' foraging trail was re-established on our apparatus, we set up recording equipment and began the experiment. We forced repeated bridge adjustments by changing the size of the gap (see Fig. 1 and Supplementary Movie 1). We increased the gap from 0 to 30 mm by 1 mm every 30 s, and then we reduced it back to 0 mm at the same rate. We chose this rate because we found it was gradual enough to avoid bridges frequently breaking, while fast enough to allow trials to be completed in a reasonable amount of time. A maximum gap size of 30 mm was a realistic choice because earlier work found naturally occurring bridges spanning gaps up to 26 mm long[24], and in another study bridges on artificial platforms grew to a maximum size of 30 mm[25]. This maximum gap

corresponds to approximately three body lengths for major workers and more than seven body lengths for some minor workers. If the bridge broke during an experiment we paused gap size increases to allow repair, terminating trials in which bridges were nonfunctional for more than 1 min. Thus, complete trials were those with sufficiently high and steady traffic to support functional bridges across the experiment. Brief breaks occurred in eight out of the ten trials, during both the expansion and contraction phases. We video-recorded experiments from both the side and the top, using Panasonic Lumix cameras (one GH3 and one GH4), recording at a minimum of 30 frames per second and a resolution of 1920 × 1080. We placed green cardstock in the background of each camera to increase contrast.

When possible, we conducted multiple trials in a row. We waited for the bridge to fully disassemble and for smooth, uninterrupted traffic to return before starting a new trial, but we did not disassemble the apparatus in between, as the process of setting up the experiments was time-consuming. Trials conducted on the same day were thus conducted with the same colony. Because colonies travel hundreds of meters through dense leaf litter and forest canopy each night, we were not able to track colony identity from one day to the next, and we cannot be sure of the total number of colonies represented in our experiments. The ten complete trials were conducted on five different days at five distinct locations—ranging from ~200 m to 1.7 km apart—and therefore represent up to five different colonies. No more than three trials were conducted on any given day, and because *E. hamatum* are relatively abundant on BCI[55] it is highly unlikely that we sampled only a single colony. *E. hamatum* colonies typically number in the hundreds of thousands of individuals, so we are confident that multiple trials were not conducted on the same individual ants.

### Data extraction

*Extraction of raw structural data.* We extracted the size and shape of the bridges from video recordings of field experiments at a temporal resolution of 1 s. As shown in Fig. 1, we first separately extracted the side-view and top-view "envelopes" (i.e., the silhouettes of the bridges projected on the 2D plane of the camera sensor), and then combined metrics about these envelopes to recreate the 3D structure of bridges, estimating volume, mean cross-sectional area, and tautness (height of the centroid of the side-view envelope). We used both Matlab (version R2017b) and R (version 3.6.3) to extract, process, and analyze these data. Full details of the structural data extraction process are included in Supplementary Note 1. Data and code are available as described in the Data and Code Availability sections, and full raw videos are available upon request.

*Extraction of raw joining and leaving data.* While we could automatically extract structural data from our recordings, tracking individual ants as they cross bridges would be impractical to attempt automatically—it is challenging even for human observers to distinguish crossing ants from the ants making up the bridge as they cross. We thus extracted data about individual decisions to join the bridge

manually. We did this using the top-view recordings. We kept track of the number of ants walking into and out of the video frame to estimate the number of ants that joined or left the bridge in a given period of time. We then used this information as a guide, carefully watching to confirm such events and find the exact times when they occurred. We confirmed that all joining and leaving events were accounted for by ensuring that the total number of joining events throughout a trial was equal to the total number of leaving events, which must be true given that each trial began and ended with zero ants in the bridge.

These data allowed us to examine the number and timing of joining and leaving events. Recording all events of ants joining or leaving the bridges also provided us with an accurate count of the number of ants in the bridge throughout trials. Combining this with structural data gave us additional metrics, such as the density of ants in the bridge (number of ants per cubic mm of the bridge).

**Models**. All three of our behavioral models (linear baseline, linear delay, and nonlinear accumulator) are probabilistic, discrete-time models implemented in R. Each timestep corresponds to a gap interval, i.e., the 30-s time period between changes in gap size. Traffic is not explicitly modeled. For each timestep, we draw the number of ants that join and leave the bridge from negative binomial distributions parameterized using the data, as further explained for each model below. We record the number of ants in each simulated bridge through time and we only allow ants to leave up to the current number of ants in the bridge (we do not allow there to be negative ants in bridges). Each simulation begins with a gap size of zero and zero ants in the bridge; as with the experiments, we model 31 expansion steps (starting with gap = 0 to gap = 30) and 30 contraction steps (starting with gap = 29 to gap = 0). The data recorded for each simulation includes the timestep, gap size, phase (expansion or contraction), the number of joining and leaving events, and the state of the bridge (number of ants).

*Parameterization of linear models*. As discussed above, the parameters for the distribution of the number of events per step were drawn from the data. We fit the distribution of the number of joining events per 30 s gap interval to a negative binomial distribution, using the fitdistrplus[56] package. We did this separately for each trial, finding the fitted mean and dispersion parameters $\mu$ and $\theta$, and used the median values of $\mu$ and $\theta$ across the trials to parameterize the baseline model. We parameterized the distribution of the number of leaving events with the same process. We excluded gap intervals during which the bridge was broken or recovering from a break, as well as a single entire trial in which the bridge was broken or recovering for the majority of the trial (trial ID "2016_03_15_trial_4").

For the linear delay model, in which we modeled a delay between ants' decisions to leave the bridge and actually leaving, we had to adjust these distributions for leaving events. For example, when modeling a delay of 1 timestep (one 30 s gap interval), we reassigned leaving events in the first step of the contraction to the expansion, for each trial, and refit the distribution. We did this for delay durations of one timestep up to six timesteps. We did not model a delay in joining or adjust the distribution of the number of joining events because the ants that are crossing the bridge move too quickly to effect a joining delay longer than a few seconds.

*Parameterization of nonlinear accumulator model*. The nonlinear accumulator model is fundamentally similar to the linear models, but instead of the likelihoods of joining or leaving being constant during each phase, they vary based on the current state of the simulated bridge. We first found the equilibrium packing density of ants in real bridges; this is the packing density (the number of ants divided by gap size) at which the probability of joining is equal to the probability of leaving. We conducted logistic regressions[57] on the probability of ants joining and leaving per gap interval at varying packing densities and found that these probabilities are equal at a packing density of 0.51 ants per mm of the gap. We assumed that this equilibrium packing density was constant across gap sizes, which is reasonable given that densities were consistent throughout trials (Fig. 2 and Supplementary Fig. 4). These regressions also confirmed that ants are more likely to join a bridge containing fewer ants than expected for the gap size (low packing density), while the inverse is true for leaving probability (Fig. 3; joining probability Binomial GLM, $\beta = -1.8$, $p < 0.0001$; leaving probability Binomial GLM, $\beta = 0.95$, $p < 0.0001$).

We then used the data to find the expected number of joining and leaving events in each timestep as a function of the deviation or difference from this equilibrium packing density (Fig. 3c). Positive difference means a deficit in the number of ants, negative indicates excess ants. Each point in the figure shows the number of events during a specific gap interval, along with the deviation at the beginning of the interval. We performed negative binomial regressions on the number of joining and leaving events as predicted by the deviation, using the MASS[58] package. These joining and leaving regressions were then fed into the model—for each timestep, we used the regressions to predict the expected number of joining and leaving events given the current deviation. The numbers of events in the simulations are then drawn from negative binomial distributions with the parameter $\mu$ equal to that expected number of events. We assumed that the dispersion parameter, $\theta$, which is also an output of the regressions, differs between the joining and leaving distributions but is constant across deviation values. We confirmed that this assumption was reasonable by analyzing the residuals of the regressions, using the DHARMa[59] package. In addition to parameterizing the

model, these regressions also confirmed that the number of joining events increases with increasing deficit (Negative Binomial GLM, $\beta = 0.049$, $p = 0.017$, $\theta = 0.773$), and the number of leaving events decreases with an increasing deficit (Negative Binomial GLM, $\beta = -0.067$, $p < 0.0001$, $\theta = 1.96$).

**Data analyses**
*Overview*. Much of our analysis of the global data associated with the collective structures involved plotting and visually examining results metrics to identify overall patterns as described in the section "Field experiments: collective structures." Additionally, we performed Pearson's product-moment correlations between the volume of bridges and the number of ants separately for each trial, to demonstrate that these metrics are tightly correlated through time, and we measured and conducted statistical analyses on the extent of hysteresis as described below. To analyze the simulation results of our competing models we plotted simulation results, compared the proportions of simulations for each model in which particular outcomes occurred, and also compared the extent of hysteresis (see the section "Mechanistic Models"). Finally, to analyze data for possible individual cues of joining and leaving, we measured and conducted statistical tests of performance and tautness, as described below.

For all analyses, we excluded data from when bridges were broken or recovering from a break, which introduces artifacts due to a build-up of ants who are unable to cross. We determined that the bridge had recovered from a particular break by performing a blind, manual examination of the plot of bridge volume over time. Estimated volume rapidly spikes when a bridge is broken as ants that are unable to cross build-up, and volume rapidly declines when recovering from a break as these extra ants cross and move on. We considered the bridge to have recovered from a break at the inflection point where the slope of the volume plot returned to a range more typical of functional bridges.

*Extent of hysteresis*. For each simulation and experimental trial, we measured the extent of hysteresis in the number of ants in the bridge as the area between the LOESS-smoothed curves for the expansion and contraction phases, normalized by the area under the higher curve; it is, therefore, a unitless proportion. This measurement is the absolute value of hysteresis; we made this value negative if the expansion curve was higher than the contraction curve on average (i.e., if the area under the expansion curve was larger). For the experimental trials, we also measured the extent of hysteresis in the volume, mean cross-sectional area, and tautness (height of the center of mass) of bridges. We performed one-sample, two-tailed $t$-tests on the extent of hysteresis in each of these four measures for the experimental data.

*Performance*. We used optical flow over bridges relative to optical flow over platforms as a measure of bridge performance, which we extracted for each 10-s interval of each trial. Details of how we produced this measure, including how we extracted optical flow from videos of experiments, are included in Fig. 5 and in Supplementary Note 2. To see if this metric is a cue influencing individual decisions to join or leave a bridge, we tested whether performance immediately before joining events differed from the performance at other times, using a resampling approach. We looked at the mean performance in the 10-s interval before joining events and compared this to the distribution of means derived from 10,000 simulated experiments in which the timing of joining events was randomized, keeping the number of events in each phase of each trial the same in these simulated tests. The resulting distributions of means during both phases are therefore null distributions that would be expected if bridge performance had no influence on when joining events occurred (Fig. 6). We analyzed the expansion and contraction phases separately, to eliminate phase as a potential confounding variable. We analyzed leaving events in the same way. To create figures from these as well as other data, we used R packages ggridges[60] and gridExtra[61]. We also confirmed that our conclusions were not sensitive to the interval length over which we measured performance with a sensitivity analysis (see Supplementary Note 2 and Supplementary Fig. 14). We used the same resampling approach to statistically analyze whether global bridge tautness influences decisions to leave a bridge.

**Reporting summary**. Further information on research design is available in the Nature Research Reporting Summary linked to this article.

## Data availability
The data[62] generated in this study have been deposited in the figshare database under accession code https://doi.org/10.6084/m9.figshare.13337255.v1.

## Code availability
The code[62] associated with this study has been deposited in the figshare database under accession code https://doi.org/10.6084/m9.figshare.13337255.v1.

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

## Acknowledgements
We thank Chris Reid and Matt Lutz for their advice and discussions regarding fieldwork and experiments. The staff of the Smithsonian Tropical Research Institute, Barro Colorado Island, Panama provided assistance and support. We thank Courtney Rockenbach, Cesar Gutierrez, and Jane McCreery for their assistance in the field. Mike Rubenstein and Melinda Malley participated in early discussions and field explorations, and we thank Mike Rubenstein for also providing comments on the manuscript. Grace Haynes performed some manual data extraction. We thank Jordan Kennedy, the Self-Organizing Systems Research Group at Harvard, and Albert Kao for helpful discussions and support. H.F.M. acknowledges support from a James S. McDonnell Postdoctoral Fellowship Award.

## Author contributions
S.G., R.N., and H.F.M. participated in study conception and experimental design. H.F.M. performed the fieldwork. S.G. and H.F.M. extracted and processed optical flow data from

videos. H.F.M. coded and performed data extraction for all other automatically extracted data. A.I.P. developed the system for extraction of manual data; A.I.P. and G.G. manually extracted these data. H.F.M. performed the data analyses and figure generation, aided by discussions with S.G. and R.N. H.F.M. developed and implemented the behavioral models. H.F.M., R.N., S.G., and G.G. participated in discussions to interpret results. H.F.M. drafted the manuscript. All authors edited the manuscript and gave approval for publication.

## Competing interests

The authors declare no competing interests.
