## [Peer Review File · Nature Communications]

Peer review comments, first round review –

Reviewer #1 (Remarks to the Author):

This study used both empirical and modeling approaches to examine how self-assembled army ant bridges cope with changing gap distances. Among many interesting and novel results, the authors report the importance of history in gap change response, but also suggest that decisions to joining and leaving bridges likely occurs via different cues. The topic of how living self assemblages cope with changing environments is interesting beyond the field of biology alone. This work has the potential to inform bio-inspired design in a variety of multi-agent systems.

Although I found this study interesting and innovative, I also had several major concerns:

1) Omitted details make assessing the soundness of conclusions difficult. Although there were numerous interesting hypotheses and comparisons within and among empirical and simulated datasets, there was little to no detail in either the main manuscript or in the supplementary methods about what statistical tests were performed to assess significance (aside from one reported Pearson correlation). Also, the authors could be more forthcoming about replicate size in terms of number of field colonies used as well as number of replicate trials within each colony. If these trials were in fact all performed using the same colony then within-colony variation is well-captured but between colony variation is poorly captured. Some would call this scenario pseudoreplication.

2) Consideration of what types of movement problems army ants typically solve while bridging in the wild (what sorts of environmental problems bridges are adapted to) is a bit lacking. The authors only go so far as to say that movement occurs. A better discussion of this could shed light on why distributed regulatory behaviors might exist as they do. I will elaborate in the in-line comments.

3) Related to this, contraction was always performed after expansion, making it hard to distinguish between strategies for dealing with a sensed closing of a gap and strategies for dealing with a perceived unstable (variable) gap distance.

Specific in-line comments:

58 & 485 – Unlike for other provided examples of army ant self-assembly, both places where bivouacs are mentioned in this manuscript cite general reviews of collective behavior but lack citations related to bivouacs specifically. Here are some suggested citations which could rectify this (Jackson 1957 used the same model species as this study):

Schneirla, T. C., Brown, R. Z., & Brown, F. C. (1954). The bivouac or temporary nest as an adaptive factor in certain terrestrial species of army ants. *Ecological Monographs*, 24(3), 269-296.

Jackson, W. B. (1957). Microclimatic patterns in the army ant bivouac. *Ecology*, 38(2), 276-285.

Baudier, K. M., D'Amelio, C. L., Sulger, E., O'Connor, M. P., & O'Donnell, S. (2019). Plastic collective endothermy in a complex animal society (army ant bivouacs: *Eciton burchellii parvispinum*). *Ecography*, 42(4), 730-739.

60 – Another potentially relevant example are just-in-time defensive “walls” army ants construct to avoid inter-colony collisions with one another. This was recently studied by Baudier & Pavlic (2020) but was also anecdotally mentioned in Daniel Kronauer's new book as well as by Carl Rettenmeyer (1963). Cues for leaving vs. joining walls appear to be different, not unlike reported for bridges in this study.

Baudier, K. M., & Pavlic, T. P. (2020). Incidental interactions among Neotropical army-ant colonies

are met with self-organized walls of ants (Hymenoptera: Formicidae). *Myrmecological News*, 30.

Kronauer, D. J. (2020). *Army Ants: Nature's Ultimate Social Hunters*. Harvard University Press.

Rettenmeyer, C. W. (1963). Behavioral studies of army ants. *estudios de comportamiento de hormigas guerreras*. *The University of Kansas Science Bulletin.*, 44(9), 281-465.

104 – Were these trials conducted using different colonies or the same one? This does not seem to be addressed in the detailed supplemental methods, but it is important for understanding the level of replication and in turn the generalizability of the results.

106 – What qualified as a “robust foraging trail”? This is only explained as “substantial traffic” in the supplementary detailed methods. A bit more detail on the trail selection process would be useful, if not here, then in SI.

Figure 2. Axes and legends are a bit too small to read comfortably. Also, image compression made it difficult to see the very small sub-panes in panels A and B.

157 – Perhaps a more ultimate hypothesis for the observation of hysteresis could lie in army ants’ environment of evolutionary adaptiveness. Exactly what kind of movement problems do army ant bridges most commonly face? Branches, leaves, understory plants, leaf litter, etc. are “mobile” objects along which army ants commonly trail. However, these substrates most typically move back-and-forth (rather than directionally) due to wind or vibration. It might be advantageous to err on the side of a bridge with too many ants when a gap shrinks (especially a shrinking gap that recently expanded) because in many cases such shrinkage is likely to be followed by a stretch of similar magnitude. I wonder if this could account for at least some of the observed differences in contracture versus expansion behavior.

187 – While I agree that this lack of correlation is suggestive, to say it is “known” seems a stretch.

189 – It would be better to specify exactly what the authors are considering the linearity of in this sentence.

365 – The most intuitive hypothesis for joining cues to me is not that overly taught bridges slow traffic, but that they distress ants in the bridge and create a break-risk. Are taught bridges more at risk of breaking?

Figure 6 – Please include units for x-axis (here and elsewhere) and provide a brief explanation of what type of performance is being discussed (ex: traffic flow). This is a fairly lengthy caption for not having included as much.

412 – I was a bit underwhelmed to find this conclusion based on that graph. Based on what has been previously said, isn’t the bridge tension always higher during the expansion phase than during the contraction phase? If so, how could the authors expect to observe similar bridge tensions during leaving events between the two? It seems a bit like a false null hypothesis. If we assume that bridges are composed of a group of ants with variable but individually consistent response thresholds for leaving in response to tension, then couldn’t this design just be showing trends for the high-RT (during the high-tension expansion phase) and low-RT individuals (during the following low-tension contraction phase) in the group rather than showing a true group-level difference between the two experimental phases?

Figure S16 – Please add units to the x-axis.

445 – Supporting this general conclusion, there is precedent for joining and leaving decisions to be based on different suites of stimuli in other army ant living structures. For instance, in as-needed walls, enemy encounter predicts joining, but both enemy encounter and nestmate collision appeared to stimulate leaving (Baudier & Pavlic 2020).

Reviewer #2 (Remarks to the Author):

In this study, McCreary et al. examine the dynamics of army ant bridge formation in response to a changing environment. Specifically, they examine ant density, tautness, and traffic flow as the gap over which the ants have formed the bridge is expanded and contracted. They identify joining and leaving events, and relate these to the measures they take. Overall, they find that ants join expanding gaps and leave shrinking gaps, but that some key measurements of the bridge depend not only the gap size, but on the direction of change – whether the gap was expanding or contracting. They conclude that they discovered a hysteresis process, wherein the bridge responds not only to the current state of the world, but also to its own previous states. They also discuss potential individual rules for joining and leaving, and conclude that different cues may trigger these two behaviors.

At the outset I must state that the computational work carried out in this study is somewhat beyond my expertise – while I can follow the logical steps the authors are taking, I cannot vouch for the validity of their methods or code, nor can I rule out that I have misunderstood some aspect of the work. As such, it is possible that my more critical comments arise due to my own misunderstanding. If so, it is incumbent upon the authors to defuse this understanding both in their response, and in the manuscript.

This is an impressive piece of work. The collection of high-quality data in the field, working with sometimes difficult subjects, is no small feat. The subject analysed (superorganismal biological structure formation) is charismatic and offers potentially easy-to-access insights into distributed decision-making in an underexplored area, and offers tantalising glimpses of applications in swarm robotics. The figures are elegant and usually easy to interpret, as is the writing.

However, I have some misgivings. My main concern is that the key evidence for hysteresis – the main finding reported – seems to be lacking. This is because the experiments were never stopped for long periods of time half-way through: expansion and contraction were continuous. To truly demonstrate hysteresis, similar experiments would need to be run, but then the bridge expansion either stopped half way through expansion and allowed to stabilise (as seems to start happening in figure S10), or expanded to full wide, allowed to stabilise, then shrunk again and stopped half way through contraction and allowed to stabilise. If these two treatments yielded different bridges, this would demonstrate true hysteresis. Otherwise, all we may be seeing is lag. I realise that the authors probably first went out to get the field data, then fished in the data for a nice story, so couldn't run such bespoke, hypothesis driven experiments. Such fishing is absolutely fine, especially in such a poorly-studied and poorly-understood system. However, it does mean that making very firm statements, as the authors do, becomes problematic.

The authors argue that bridges should respond so quickly as to not show lag based on previous findings (line 155), but this is unconvincing and a different situation which is not directly relevant here. The authors do attempt to address this lag vs hysteresis issue using a modelling approach, by simulating what a 'lag' system would produce (linear model with delay). However, I am afraid I was unconvinced by the appropriateness of the linear models. Specifically, the models respond directly to the phase of the experiment (expansion or contraction). This makes little biological sense – how could the ants in the bridge know the phase? As such, I found these linear models 'straw men' to be conveniently knocked down, rather rigorous alternative hypotheses. The accumulator model is much more realistic, in that it reads the state of the bridge, not the phase of the experiment. Nicely, it does a much better job of reproducing the empirical data. However, this model could and should have been taken further. For example, the authors could simulate the explicit test for hysteresis I suggested above using their data, and see if hysteresis really emerges. They also could add a lag to the additive model, and see if this reproduces the empirical behavior even better.

On a related note, I would have liked more clarity of the hypothesised mechanism driving hysteresis. It seems reasonable that if joining effects require lower threshold deviations than leaving effects, this could result in the hysteresis described, but (unless I missed it) this was never explicitly stated, nor presented graphically via a bare-bones mathematical model, perhaps populated with real values from the data.

Also – the title implies that hysteresis stabilises the dynamic control. Well, it does in other systems, but where is the evidence that it does here? Did I miss this? Indeed, I felt like in fact the authors were implying that the bridges WERE under constant adjustment, with ants leaving and joining due to different reasons, but resulting in a constant bridge density. This seems to contradict line 37.

More broadly, while the work is charismatic and elegantly carried out, I don't feel like any deeper insights are gained into collective building by social insects, nor are any broader insights gained about collective behavior. The authors write "Our work provides general insight into the dynamic control of distributed systems", but this is an empty statement: what, specifically, have we learned about dynamic control of distributed systems that we didn't know before?

More convincing was the elucidation of the behavioral rules governing joining and leaving. While I have some quibbles over the specific things the ants are sensing (see minor comments), the finding that joining and leaving are driven by different things seems quite robust, and is quite interesting. This is not explored very deeply, but may have interesting adaptive repercussions. I also agree that this system can result in hysteresis, if these two responses have different effective thresholds – but this needs spelling out better.

In conclusion, this study offers one of the few attempts to examine an interesting, charismatic, and potentially broadly applicable topic of dynamic response of superorganismic structures. I have no doubt that it would be read with interest by a broad range of researchers, and be of interest to the general public as well. However, I am not convinced that the main conclusions are supported by evidence, and have other misgivings as well (see minor comments below). I am thus not convinced that anything really new has been firmly demonstrated (hysteresis in collective behavior has been shown before), nor that any large insights have been gained. If the authors could assuage my doubts on these points, I would find this work to be a good fit for Nature Communications. If not, then not.

Minor comments

- Introduction: I was surprised that no mention was made of other collective behaviors in social insects, nor the frameworks used to understand them. Do the theoretical frameworks used to understand collective transport, or collective, dynamic choice of food sites, also offer a framework for this collective behavior?
- Line 72 – 'ground' isn't quite right. Substrate is better, but I admit, less catchy.
- I didn't find paragraph 79-91 very enlightening, as I really only understood them having read the manuscript and supplement.
- Ln 94 here, for traditionalists who like to know what was actually done before they read the interpretation, please specifically state that the detailed methods are found in S1.
- Ln 128 – F – odd that droop is not maximum at the maximum extension, but rather at middle 15-20. This doesn't seem to be explained anywhere, nor even mentioned.
- Figure 3B – perhaps removing the extreme values in this figure (anything above 2 or 4) would make this a more useful figure? Similarly, the Y axis of C could be trimmed a bit to zoom in on the action.
- Lines 176-178 – nice, and interesting, but this is just a story unless there are data to back it up.
- Line 188 – no it isn't: there are clear dips at the start and starting at c. 20 seconds. Why is that? Is this a sampling artefact? See later comment as well.
- Line 231 – please explain briefly why.
- Line 290 – sorry, what was non-linear about this feedback?
- Line 337 – 339 or that a slack bridge affects your measurements somehow. Really, I don't see why flow was not measured directly - the traffic is not THAT high, and you hardly get any seriously overlapping ants. And even if you did, you could easily get the real count from either the top or the side. Inferring by proxy and software opens the door to lots of unexpected biases, and probably didn't save a lot of time in the end.
- Figure 5(2) – Dot fill is not helpful. I can tell which side of the line each dot is on even without fill. Rather, you could use the fill for something more informative, like to see expansion vs contraction phase. 'I' is also missing in 'brdge' on Y axis
- Lines 384-5 – probably bridges with redundancy are more resilient to individual ants losing their grip. You could maybe test this: is a break more likely to happen on an expansion in sparser bridges?
- Lines 431-439 – I am unconvinced that the tautness of the bridge is in any meaningful way

affected by the ant traffic. Firstly, by simply examining the video we can see that the bridge doesn't change at all when ants step on and off it. Secondly, the weight of a few ants compared to the strength of the ants is so minimal that we should probably not expect to see an effect.

- Supplement – key information missing: how many colonies? How many trails? How long a pause between trials? Any reason to think that some chemical cues are left on the platform (eg a 'build bridge here' pheromone) which changes the behavior from first trial to second? What was the width of the bridge?

- Sup – Cardstock should be mentioned in the field experiments bit

- Sup – "For short periods of time (~30 seconds, depending..." a bit unclear: you went through the whole of all the videos like this, right?

- Sup – "For each time step, we draw the number of ants that join and leave the bridge from normal distributions with the same mean and standard deviation as the data, as further explained for each model below."

Did the original data have a normal distribution? Why not draw it from the data directly? Or at least a model of the data?

- Sup – "recovering from a break" how was this defined objectively?

- S10 – This data is a bit puzzling. Wouldn't we expect this to have a poisson-shaped distribution? Why does the number of events slip down towards the 30 second mark? Did you simply get lucky with 30 seconds being more or less how long ants need to recover? Or is this something about preparing to move bridge (e.g. coming closer to it, putting hand on cog, etc)?

- S14 – what are the units of the Y axis? State explicitly what the circles are.

Reviewer #3 (Remarks to the Author):

The authors present a study of army ant bridges. I liked the experimental part of the work, where they lengthened and contracted gaps underneath the bridge, and showed that army ants exhibit hysteresis: they retain more ants during contraction than during expansion. These dynamic tests are very original, and reveal a new feature of ant rafts that is in common with many materials. I think the experimental work is publishable on its own and would make a good paper.

The experimental work is well described, but the theoretical exposition could be greatly improved. Despite reading quite a lengthy text, I am not convinced by the title of the work, stating that hysteresis is stabilizing for the rafts. I would have liked to see more details on how the various models were tested. The figures are too many and not clearly presented. The flow charts in the figures are difficult to read and I would have preferred equations. The parameters used in each model should be presented more clearly.

line 70 What kind of things will disturb twigs and leaves while ants cross over them? Wind? Animals? What is the duration that a bridge needs to stay in position?

102 The largest bridge in this study is composed of only 19 ants. It would be useful if this number were stated in the abstract to give the reader an idea of the scale involved.

104 The authors state the dynamics are repeatable, but the figure shows high variability. Can they explain what the Pearson correlation says in this case?

Fig 1. The authors conduct experiments to a maximum gap of 30 mm. What are these units in ant body lengths? Why was the contraction and expansion rate conducted at the rates performed? Similarly what is the rationale for making the gap this length?

Fig 2. The caption has a typo: "results metrics"

This figure suggests that only 10 trials were conducted and that at least 6 of the trials had breaks that might influence the data. Do the authors' measurements of hysteresis still hold for the bridges that did not break? Did the breaks happen more in the extension or the contraction? More comment should be given here besides the caption.

Parts A and B are very difficult to read, especially the insets. A different color combination might be used.

165 This paragraph on hysteresis is very nice, but I think it should be in introduction. The authors might also note that there are few theoretical models that can explain hysteresis.

194 the authors should explain the data-driven rates of joining and leaving. Are these numbers reasonable? It seems like the numbers are 1-2 ants per minute.

Fig 3. The authors should clearly derive and explain each of these models. The font is quite blurry and the equations are difficult to read. I am not clear how these models describe the dynamics on the bridge.

Fig 3 units should be clarified. Is mean rate per minute in terms of number of ants?

200 I fundamentally do not understand the behavior for leaving and joining the model. I understand there is traffic of near constant flow rate. Do ants make decisions as they cross the bridge? Do all ants in the bridge make decisions at each time steps? How are these decisions made? Using a random variable?

Some of these questions are addressed in the Supplement, but I would have liked more numbers given in the text. It sounds like the rates of joining and leaving are taken from a normal distribution. The authors should still state what are the factors that lead to these numbers.

The Table in the supplement is not labelled. It also has no units.

248 The nonlinear accumulator model should be renamed because it is simply not specific enough. There are an infinite number of accumulator models, and this one is proportional to the excess or deficit ants in the bridge. The title should reflect that.

There is much reliance on the supplement, which makes the paper difficult to read. The authors refer the reader to the burstiness in the Supplement.

Figures 4-6 give three new variables: the extent of hysteresis, the performance metric, and the flow over bridge and panels. The units for the latter two are not given in the figure. These variables should also be clearly defined.

Overall, the paper has some interesting experimental results, but the theoretical work could have been better explained. It seems like there are some negative results here (linear models don't work). The accumulator model seems to have more promise, but it is qualitatively matching at best. The paper also suffers from length and over-reliance on the supplement, and many variables and measurements not clearly defined. Some of these are due to the constraints of the word limit for this journal. This paper may benefit from being split into several publishable pieces.

Reviewer #4 (Remarks to the Author):

This paper investigates bridge formation in *Eciton hamatum* army ants. The authors present experimental and simulation results that demonstrate the presence of hysteresis in the structural properties of the bridges as the size of the gap they span is increased and decreased. Overall this is an interesting paper that I believe could be suitable for a general interest journal however I do think the authors need to improve the manuscript. In general, I found the manuscript to be quite long and not as focused on the key findings as I would expect for this journal. I believe the authors could improve the paper in this regard and it would find a broader audience. I have several other comments which can be found below.

1. The motivation for the study in terms of the adaptive role of the observed collective behavior is fairly weak. The authors claim that coping with gaps that grow and shrink gradually "occurs frequently because they are built on twigs, leaves, and other unstable features". It's hard to imagine how the real challenges the ants face (in terms of unstable features) maps to the experimental set-up of a steadily increasing and reducing gap. I would like to see a much better justification that the observed hysteresis is biologically relevant or a case made that this investigation has relevance in other contexts, i.e. that the presence of structural memory is important even if the exact conditions simulated are not present in the natural environment.

2. I don't believe the issue of whether the speed at which the gap is increased or decreased is properly addressed. The hysteresis could well be a transient phenomena that would disappear if the rate at which the gap changed was decreased. While I don't expect the authors to rerun experiments, I would like to see this investigated in the simulations. It wouldn't be too difficult to show that there are genuinely alternate stable states in the simulation model for a fixed gap size i.e. if the increase/decrease of the gap was stopped midway would an increasing trajectory reach a stable steady state that was different to a decreasing trajectory? I don't believe that whether the hysteresis is rate dependent or not affects the novelty of the study but it should be explored in more detail.

Minor comments

- I don't think the term 'dynamic control' would be well understood outside the engineering community. I would consider rewording the title and the abstract to make the paper more accessible.

L79 - I found this framing as a sequence of questions a bit odd - the 'how' of question 1 would seem to encompass questions 2 and 3, while the initial "Do the collective structures adapt.." seems redundant - we're already told they do on L68. I would much prefer if this was framed in terms of experimental results followed by simulation study.

- I'm curious as to the role of bridges breaking in the process. In both panels in the figure there is a shaded region indicating a break and recovery. Presumably this only happens in the expansion phase, but does this always happen? Is there any relation between the occurrence of a break and the hysteresis volume?

- the figure insets are very small - I'm not sure they add much value.

L178 - the authors state "In army ant bridges, hysteresis may play a similar role, improving stability and preventing ants from constantly joining and leaving when the bridge is close to its optimal size." This is an interesting statement but it's not clear exactly how this would work. Could this be explained in more detail? Is this referring to a scenario when the gap is contracting or static? How would the hysteresis prevent ants from constantly joining/leaving, doesn't the hysteresis emerge from the decisions of the ants?

I'm not convinced the inclusion of the linear models in the main text is necessary, especially as one model is included only for completeness (L202). The nonlinear model is the only one really of interest given the results so I would relegate the other models to the SI and include only to demonstrate that the nonlinearity is required.

L406 to L478 - this section is quite long and comes across as fairly speculative. Consider condensing to the a few key points.

L429 a reference is required here

For the following, reviewers' comments are in black, and our responses are in blue.

Reviewer #1 (Remarks to the Author):

This study used both empirical and modeling approaches to examine how self-assembled army ant bridges cope with changing gap distances. Among many interesting and novel results, the authors report the importance of history in gap change response, but also suggest that decisions to joining and leaving bridges likely occurs via different cues. The topic of how living self assemblages cope with changing environments is interesting beyond the field of biology alone. This work has the potential to inform bio-inspired design in a variety of multi-agent systems.

We thank Reviewer 1 for their thoughtful comments, which helped us considerably improve the manuscript.

Although I found this study interesting and innovative, I also had several major concerns:

1) Omitted details make assessing the soundness of conclusions difficult. Although there were numerous interesting hypotheses and comparisons within and among empirical and simulated datasets, there was little to no detail in either the main manuscript or in the supplementary methods about what statistical tests were performed to assess significance (aside from one reported Pearson correlation). Also, the authors could be more forthcoming about replicate size in terms of number of field colonies used as well as number of replicate trials within each colony. If these trials were in fact all performed using the same colony then within-colony variation is well-captured but between colony variation is poorly captured. Some would call this scenario pseudoreplication.

We have added additional explanations and support material for our statistical tests (see lines 154-161, 657-664, 669-681, and 683-729). For many of our analyses, traditional approaches which produce p-values were inappropriate because their assumptions were violated—in particular, our data have temporal autocorrelation and are not normally distributed. We therefore used other approaches, such as resampling analyses which allow one to use the data itself to develop an expected distribution of the response variable under the null hypothesis (described in Wilcox 2003 and other textbooks). This avoids making any parametric assumptions about how the data are distributed. While we don't provide exact p values, we still report whether our results were significant at the familiar $\alpha = 0.05$ level, by determining whether our actual results are more or less extreme than the most extreme 5% of that null distribution.

We have also added more detailed information to the methods regarding the number of colonies and replicate trials within colonies, and we thank Reviewer 1 for pointing out this omission. Due to the fact that *Eciton hamatum* colonies are nomadic and migrate through dense leaf litter and forest canopy, we were not able to track colony identity from one day to the next. Therefore, as we now explain in lines 586-595, we cannot be sure that we did not conduct trials with the same colony on multiple days, or of the total number of distinct colonies. Nevertheless, we are confident that multiple trials were *not* conducted on the same individuals, because of the large size of colonies. We conducted the ten complete trials on five different days at five distinct locations—ranging from approximately 200 m to 1.7 km apart. Therefore up to five distinct colonies are represented. No more than three trials were conducted

on any given day. *E. hamatum* ants are relatively abundant on BCI (O'Donnell et al. 2007), so it is highly unlikely that only a single colony is represented by our data.

We also note that while quantitative differences among social insect colonies in various behaviors are common (Horna-Lowell et al. 2021), we do not expect to see substantial *qualitative* differences (Walsh et al. 2020), nor do we observe such differences in our study. Behavioral patterns in all of our trials were similar, and trials conducted on the same day were not qualitatively more similar than those conducted on different days. The difficulty of tracking colony identity is common in army ant research—especially research conducted on foraging trails—and several previous published studies have also either been unable to specify an exact number of colonies and/or have apparently assumed that different raiding columns represent different colonies (including Powell & Franks 2007, Reid et al. 2015, and Lutz et al. 2021). We agree that this is a limitation, yet we think that army ant bridges are well worth studying despite this challenge. Given the previously published literature with similar limitations, it is apparent that other researchers and journals agree. Nevertheless, we should have been clearer about this limitation in our manuscript as Reviewer 1 noted; we have addressed this in the revised version (lines 586-595).

Horna-Lowell, E., Neumann, K. M., O'Fallon, S., Rubio, A. & Pinter-Wollman, N. Personality of ant colonies (Hymenoptera: Formicidae) – underlying mechanisms and ecological consequences. *Myrmecological News* **31**, (2021).

Lutz, M. J. *et al.* Individual error correction drives responsive self-assembly of army ant scaffolds. *PNAS* **118**, (2021).

O'donnell, S., Lattke, J., Powell, S. & Kaspari, M. Army ants in four forests: geographic variation in raid rates and species composition. *Journal of Animal Ecology* **76**, 580–589 (2007).

Powell, S. & Franks, N. R. How a few help all: living pothole plugs speed prey delivery in the army ant *Eciton burchellii*. *Animal Behaviour* **73**, 1067–1076 (2007).

Reid, C. R. *et al.* Army ants dynamically adjust living bridges in response to a cost–benefit trade-off. *PNAS* 201512241 (2015) doi:[10.1073/pnas.1512241112](https://doi.org/10.1073/pnas.1512241112).

Walsh, J. T., Garnier, S. & Linksvayer, T. A. Ant Collective Behavior Is Heritable and Shaped by Selection. *The American Naturalist* **196**, 541–554 (2020).

Wilcox, R. R. *Applying Contemporary Statistical Techniques*. (Elsevier, 2003).

2) Consideration of what types of movement problems army ants typically solve while bridging in the wild (what sorts of environmental problems bridges are adapted to) is a bit lacking. The authors only go so far as to say that movement occurs. A better discussion of this could shed light on why distributed regulatory behaviors might exist as they do. I will elaborate in the in-line comments.

We agree that we should have provided more natural context for the behavior we study. We have added a better discussion of this, as well as an extension of our model, as explained below, in response to the comment referencing line 157. We think our study is much stronger because of these additions, and we really appreciate Reviewer #1's thoughtful attention to this.

3) Related to this, contraction was always performed after expansion, making it hard to distinguish between strategies for dealing with a sensed closing of a gap and strategies for dealing with a perceived unstable (variable) gap distance.

While we could not perform additional field experiments, our revised manuscript includes some model extensions testing variable gaps (see response to the comment referencing line 157).

Specific in-line comments:

58 & 485 – Unlike for other provided examples of army ant self-assembly, both places where bivouacs are mentioned in this manuscript cite general reviews of collective behavior but lack citations related to bivouacs specifically. Here are some suggested citations which could rectify this (Jackson 1957 used the same model species as this study):

Schneirla, T. C., Brown, R. Z., & Brown, F. C. (1954). The bivouac or temporary nest as an adaptive factor in certain terrestrial species of army ants. *Ecological Monographs*, 24(3), 269-296.

Jackson, W. B. (1957). Microclimatic patterns in the army ant bivouac. *Ecology*, 38(2), 276-285.

Baudier, K. M., D'Amelio, C. L., Sulger, E., O'Connor, M. P., & O'Donnell, S. (2019). Plastic collective endothermy in a complex animal society (army ant bivouacs: *Eciton burchellii parvispinum*). *Ecography*, 42(4), 730-739.

We thank Reviewer 1 for pointing out this inadvertent omission. We have added these citations in the introduction (line 52).

60 – Another potentially relevant example are just-in-time defensive “walls” army ants construct to avoid inter-colony collisions with one another. This was recently studied by Baudier & Pavlic (2020) but was also anecdotally mentioned in Daniel Kronauer’s new book as well as by Carl Rettenmeyer (1963). Cues for leaving vs. joining walls appear to be different, not unlike reported for bridges in this study.

Baudier, K. M., & Pavlic, T. P. (2020). Incidental interactions among Neotropical army-ant colonies are met with self-organized walls of ants (Hymenoptera: Formicidae). *Myrmecological News*, 30.

Kronauer, D. J. (2020). *Army Ants: Nature's Ultimate Social Hunters*. Harvard University Press.

Rettenmeyer, C. W. (1963). Behavioral studies of army ants. *estudios de compor-tamiento de hormigas guerreras*. The University of Kansas Science Bulletin., 44(9), 281-465.

We have added these citations as suggested (see lines 55-56), and have also included more discussion of this behavior as suggested below.

104 – Were these trials conducted using different colonies or the same one? This does not seem to be addressed in the detailed supplemental methods, but it is important for understanding the level of replication and in turn the generalizability of the results.

As explained above, due to the impracticality of tracking colonies in the field, trials from a single day were conducted on the same colony, and we cannot be sure that trials on separate days were conducted on different colonies. We have added this information to the methods section as well as an explanation (see lines 583-595).

106 – What qualified as a “robust foraging trail”? This is only explained as “substantial traffic” in the supplementary detailed methods. A bit more detail on the trail selection process would be useful, if not here, then in SI.

We have added these details to the methods (see lines 545-548). Specifically, because bridges tend to disassemble when traffic drops, we chose foraging trails with high traffic that seemed to be relatively consistent, to improve the chance that traffic would be maintained long enough to complete a trial.

Figure 2. Axes and legends are a bit too small to read comfortably. Also, image compression made it difficult to see the very small sub-panes in panels A and B.

We thank Reviewer 1 for pointing this out. We have improved the readability of this figure by increasing the font size for the axes and legend. We have also reduced the number of sub-panes to three, so that they can be larger, the trials that are now excluded are included in the supplement. Similarly, we reduced the number of sub-panes in figures showing model results (formerly part of figure 3, now part of figure 4).

157 – Perhaps a more ultimate hypothesis for the observation of hysteresis could lie in army ants’ environment of evolutionary adaptiveness. Exactly what kind of movement problems do army ant bridges most commonly face? Branches, leaves, understory plants, leaf litter, etc. are “mobile” objects along which army ants commonly trail. However, these substrates most typically move back-and-forth (rather than directionally) due to wind or vibration. It might be advantageous to err on the side of a bridge with too many ants when a gap shrinks (especially a shrinking gap that recently expanded) because in many cases such shrinkage is likely to be followed by a stretch of similar magnitude. I wonder if this could account for at least some of the observed differences in contracture versus expansion behavior.

We thank Reviewer 1 for this suggestion, along with the related comment #2. As these comments point out, we did not sufficiently put our results into the context of the ecology and evolution of natural ant colonies. We have added context to the introduction and discussion to address this (see lines 68-73 and 526-537). Furthermore, we now test our models under a new experimental condition of small, back-and-forth movement, which we agree is likely common for these structures. Briefly, our results show that hysteresis in the accumulator model maintains stable, viable bridges under these conditions, while the linear models produce bridges which quickly become unstable (lines 283-298, 342-354). We

found the results from this really interesting, and we thank Reviewer 1 for sparking this model extension.

187 – While I agree that this lack of correlation is suggestive, to say it is “known” seems a stretch.

We have changed this sentence, it now reads “... joining and leaving events do not appear to be directly prompted by changes in gap size...” (line 197).

189 – It would be better to specify exactly what the authors are considering the linearity of in this sentence.

Thank you for pointing this out. We have revised the sentence to clarify this; it now reads “... three behavioral models of individual decisions to join or leave structures—two linear with respect to the gap perturbation and one nonlinear...” (line 195). We also clarify in line 333 that the accumulator model is nonlinear because the average number of joining or leaving events in each timestep (μ) depends nonlinearly on the deficit/excess ($\Delta(t)$) as shown in Figure 3C.

365 – The most intuitive hypothesis for joining cues to me is not that overly taugh bridges slow traffic, but that they distress ants in the bridge and create a break-risk. Are taugh bridges more at risk of breaking?

This is an interesting point, and we agree that it would be worthwhile to rigorously examine how individuals’ behavior affects the risk of breaks. We did not do so in this paper because we were focusing on complete trials, which were those in which traffic was steady enough to have only brief breaks, if any, and therefore contained only a small number.

It indeed seems likely that additional ants may strengthen bridges, regardless of how traffic is affected. We now mention this possibility (see line 456), and we thank Reviewer #1 for the suggestion. We do not think this satisfactorily explains why ants would be *more* likely to join high performance bridges than low (with our measure of performance), however, and so we still discuss the possibility that this result reflects ants investing in portions of the trail that are functioning well.

Figure 6 – Please include units for x-axis (here and elsewhere) and provide a brief explanation of what type of performance is being discussed (ex: traffic flow). This is a fairly lengthy caption for not having included as much.

We now provide an explanation of the performance metric in the caption (line 426). Our measure of performance is the residuals from regressions of optic flow (pixels per video frame) over bridges on optic flow over platforms. The units are therefore quite non-intuitive. We do not think adding the units to the figure would provide any additional benefit beyond the added explanation in the caption, but are happy to add them at the editor’s discretion.

412 – I was a bit underwhelmed to find this conclusion based on that graph. Based on what has been previously said, isn’t the bridge tension always higher during the expansion phase than during the contraction phase? If so,

how could the authors expect to observe similar bridge tensions during leaving events between the two? It seems a bit like a false null hypothesis. If we assume that bridges are composed of a group of ants with variable but individually consistent response thresholds for leaving in response to tension, then couldn't this design just be showing trends for the high-RT (during the high-tension expansion phase) and low-RT individuals (during the following low-tension contraction phase) in the group rather than showing a true group-level difference between the two experimental phases?

To make sure we are interpreting this comment correctly we will first rephrase it, and if we have misunderstood we would be happy to try again. We understand Reviewer 1's point to be that we may have the following issue related to survivorship bias in comparing the two experimental phases: In the expansion phase, when bridge tautness is high on average, only ants that have a *high leaving tension threshold* will leave the bridge. Then, during the contraction phase when tautness is low on average, these high-threshold ants have already left, and the remaining ants have a *low leaving tension threshold*. So, if we compare the two phases we are actually comparing two different populations, and we cannot speak to any group-level difference among phases.

However, in fact we are not attempting to compare these two phases, and we are not comparing these two hypothetical populations of ants. Instead, our comparisons are *within* each phase. We show that regardless of the typical tautness of a phase (high in expansion, low in contraction), the tautness measured before leaving events is not significantly different from tautness measurements done at random points. In other words, tautness is not predictive of ants leaving the structure, whether average tautness is high (expansion phase) or low (contraction phase). We plot this separately for each phase because phase is acting like a random effect in this analysis. We have clarified this in the text (lines 471 and 720-724) and in the caption for SI Figure S17.

We have an additional point related to this. We were able to analyze the tautness of a bridge (a global measure), but this is not the same as the tension experienced by individual ants (a local measure that we don't have). While we expect more ants to experience high tension in a taut bridge on average, these variables are not always directly related. For instance, even in a very slack bridge ants acting as anchors on the edge may experience high local tension due to the weight of the bridge itself. Likewise, in a very taut bridge there may be ants experiencing low tension due to their position in the structure. This is one reason that while we did not see an effect of global tautness on leaving, we still find it reasonable to hypothesize that local tension may prompt leaving (discussed in lines 472-479). We have further clarified this in line 476.

Figure S16 – Please add units to the x-axis.

The x axis is unitless, as it is centered tautness (the z score). This should have been clearer, and we have clarified it in the caption and by changing the axis title to “mean tautness before leaving events (z score).”

445 – Supporting this general conclusion, there is precedent for joining and leaving decisions to be based on different suites of stimuli in other army ant living structures. For instance, in as-needed walls, enemy encounter predicts joining, but both enemy encounter and nestmate collision appeared to stimulate leaving (Baudier & Pavlic 2020).

We thank Reviewer 1 for pointing this out, and we have added this information, including the noted citation (see line 499).

Reviewer #2 (Remarks to the Author):

In this study, McCreary et al. examine the dynamics of army ant bridge formation in response to a changing environment. Specifically, they examine ant density, tautness, and traffic flow as the gap over which the ants have formed the bridge is expanded and contracted. They identify joining and leaving events, and relate these to the measures they take. Overall, they find that ants join expanding gaps and leave shrinking gaps, but that some key measurements of the bridge depend not only the gap size, but on the direction of change – whether the gap was expanding or contracting. They conclude that they discovered a hysteresis process, wherein the bridge responds not only to the current state of the world, but also to its own previous states. They also discuss potential individual rules for joining and leaving, and conclude that different cues may trigger these two behaviors.

At the outset I must state that the computational work carried out in this study is somewhat beyond my expertise – while I can follow the logical steps the authors are taking, I cannot vouch for the validity of their methods or code, nor can I rule out that I have misunderstood some aspect of the work. As such, it is possible that my more critical comments arise due to my own misunderstanding. If so, it is incumbent upon the authors to defuse this understanding both in their response, and in the manuscript.

This is an impressive piece of work. The collection of high-quality data in the field, working with sometimes difficult subjects, is no small feat. The subject analysed (superorganismal biological structure formation) is charismatic and offers potentially easy-to-access insights into distributed decision-making in an underexplored area, and offers tantalising glimpses of applications in swarm robotics. The figures are elegant and usually easy to interpret, as is the writing.

However, I have some misgivings. My main concern is that the key evidence for hysteresis – the main finding reported – seems to be lacking. This is because the experiments were never stopped for long periods of time half-way through: expansion and contraction were continuous. To truly demonstrate hysteresis, similar experiments would need to be run, but then the bridge expansion either stopped half way through expansion and allowed to stabilise (as seems to start happening in figure S10), or expanded to full wide, allowed to stabilise, then shrunk again and stopped half way through contraction and allowed to stabilise. If these two treatments yielded different bridges, this would demonstrate true hysteresis. Otherwise, all we may be seeing is lag. I realise that the authors probably first went out to get the field data, then fished in the data for a nice story, so couldn't run such bespoke, hypothesis driven experiments. Such fishing is absolutely fine, especially in such a poorly-studied and poorly-understood system. However, it does mean that making very firm statements, as the authors do, becomes problematic.

We thank Reviewer 2 for this comment. We have expanded and clarified our explanations of hysteresis and our models in our revised manuscript, as described in response to the following comments.

Our goal was to study how army ant bridges respond to dynamic gaps that change in size rather than bridges at steady-state with respect to the geometry of the terrain because this had not previously been explored. The reviewer suggests that if we had stopped the experiments for a long time half-way through, then bridges in each phase would eventually reach the same state—we agree that this is possible, though not very likely (as we will discuss in the next paragraph and in response to the next comment). In fact, we *do* expect the extent of hysteresis to depend on the rate of gap-size change, as we now show in our new rate sensitivity analysis of the accumulator model. The hysteresis may be rate-dependent, yet this is still considered hysteresis. Many hysteretic systems have this kind of rate dependency, and even systems for which the mechanism is a simple lag in response are still considered to have hysteresis. A common definition of hysteresis is simply that the state of a system lags behind changes in the effect causing it. Even if hysteresis in the army ant bridges were caused by a lag, we would still find its presence interesting and relevant for this system.

Yet as discussed in our manuscript (lines 167-171) we find it unlikely that a lag is a sufficient explanation for the hysteresis we observe. Earlier work has demonstrated that ants can effect change in a matter of seconds (Garnier, et al. 2013). Our delay model, which incorporates a lag in ants leaving the structures, performed poorly regardless of the length of delay. Furthermore, the bridges we observed often remained stable, with no ants joining or leaving, through several changes in gap size before adjusting, as predicted by our accumulator model—if there were simply a lag in the systems response, the bridges would have a delayed adjustment to *every* change.

The authors argue that bridges should respond so quickly as to not show lag based on previous findings (line 155), but this is unconvincing and a different situation which is not directly relevant here. The authors do attempt to address this lag vs hysteresis issue using a modelling approach, by simulating what a ‘lag’ system would produce (linear model with delay). However, I am afraid I was unconvinced by the appropriateness of the linear models. Specifically, the models respond directly to the phase of the experiment (expansion or contraction). This makes little biological sense – how could the ants in the bridge know the phase? As such, I found these linear models ‘straw men’ to be conveniently knocked down, rather rigorous alternative hypotheses. The accumulator model is much more realistic, in that it reads the state of the bridge, not the phase of the experiment. Nicely, it does a much better job of reproducing the empirical data. However, this model could and should have been taken further. For example, the authors could simulate the explicit test for hysteresis I suggested above using their data, and see if hysteresis really emerges. They also could add a lag to the additive model, and see if this reproduces the empirical behavior even better.

We have added several extensions of our models to our study (see lines 201-204, 283-298, and 342-361, also briefly described in the last paragraph of this response). We now test all three models under conditions of small, momentary shifts in terrain (vibration-like movement), as well as conducting scalability and sensitivity tests of the accumulator model. We thank Reviewer 2 for this suggestion, which has substantially improved the manuscript.

Regarding our inclusion of the linear models, we agree that ants likely cannot directly sense the experimental phase. But it does seem reasonable that they may sense the gap-size perturbations by feeling movement—e.g sensing being more or less stretched depending on whether the gap size was increased or decreased, etc. The linear models represent reasonable, simple hypotheses for mechanisms of bridge formation—it is reasonable to expect the system's response to be proportional to the perturbation, and reasonable to expect a delay in ants leaving the bridge. It would be inappropriate to

present a model with a more complex mechanism (the accumulator model), without showing that such complexity is justified.

Furthermore, the linear models are even more important in the revised manuscript, because of the new results for the models' behavior under vibration-like conditions. The accumulator model maintains viable bridges during these vibrations, while the linear models do not—producing bridges which gradually disassemble even with a large, but vibrating, gap. This contrast demonstrates the value of the hysteresis produced by the accumulator model for stability. We have revised our model descriptions and discussion to clarify the rationale for the linear models, as well as what we learn from them (lines 298-308).

On a related note, I would have liked more clarity of the hypothesised mechanism driving hysteresis. It seems reasonable that if joining effects require lower threshold deviations than leaving effects, this could result in the hysteresis described, but (unless I missed it) this was never explicitly stated, nor presented graphically via a bare-bones mathematical model, perhaps populated with real values from the data.

If our interpretation of this comment is correct, it seems Reviewer 2 is saying that another mechanistic model we could have tested would assume that joining and leaving decisions are based on the deviation in the number of ants from the equilibrium, just as in our accumulator model, but with specific, distinct thresholds for deviations that would lead to joining and leaving events. For example, ants might *join* bridges once the deficit surpasses 2, but only *leave* once the excess surpasses 3.

We agree that such a model may result in hysteresis similar to what we observed in the real structures. Actually, this model is consistent with our accumulator model, and in a sense is a more specific version of it. The only difference between the two is that in the accumulator model we do not assume that there is a threshold structure to the ants' decisions, instead we take the probability of joining or leaving at different deviations directly from the data. We wanted to avoid imposing additional assumptions about how the deviation affects ants' decisions, and to the extent possible, let the data speak for itself. The accumulator model predicts hysteresis with data-driven individual decisions; it is not necessary to apply explicit thresholds. We are happy to revisit this issue if the reviewers or editor feels there is reason to do so, or if we have misinterpreted the comment.

Also – the title implies that hysteresis stabilises the dynamic control. Well, it does in other systems, but where is the evidence that it does here? Did I miss this? Indeed, I felt like in fact the authors were implying that the bridges WERE under constant adjustment, with ants leaving and joining due to different reasons, but resulting in a constant bridge density. This seems to contradict line 37.

Regarding line 37, we did not mean to suggest that there were not frequent adjustments in our experiment, but rather that hysteresis prevents real bridges from constantly adjusting when they are already at or near the right size. This is clearer in the revised manuscript. The statement itself has been removed because we shortened the abstract to fit within the word limit, and we have included substantially more context for the idea of stabilizing hysteresis, and the consequences of hysteresis in general.

Reviewer 2 is correct that our original manuscript included only indirect evidence that the hysteresis we observed stabilizes these structures. However, the model extensions that we have added to the manuscript, sparked by reviewer suggestions, in fact provide such direct evidence of stability.

Specifically, hysteresis allows the accumulator model to maintain viable bridges with a consistent number of ants under simulated vibration-like conditions (see above). We discuss this explicitly in the revised manuscript (line 347-354).

More broadly, while the work is charismatic and elegantly carried out, I don't feel like any deeper insights are gained into collective building by social insects, nor are any broader insights gained about collective behavior. The authors write "Our work provides general insight into the dynamic control of distributed systems", but this is an empty statement: what, specifically, have we learned about dynamic control of distributed systems that we didn't know before?

We agree that our statement about the general insight provided by the study was too vague. We have modified this statement to make clear the specific broader insights that our work provides (line 537-541). Our work is the first to demonstrate decentralized, dynamic control and hysteresis in a self-assembled structure in social insects. To our knowledge, this had also not previously been formally demonstrated in other forms of social insect construction either (e.g. underground nests, termite mounds). More broadly, by illuminating the presence and benefit of stabilizing hysteresis in this system we demonstrate the potential of related hysteretic mechanisms for stable control of decentralized systems, which may operate in other biological or engineered systems as well.

More convincing was the elucidation of the behavioral rules governing joining and leaving. While I have some quibbles over the specific things the ants are sensing (see minor comments), the finding that joining and leaving are driven by different things seems quite robust, and is quite interesting. This is not explored very deeply, but may have interesting adaptive repercussions. I also agree that this system can result in hysteresis, if these two responses have different effective thresholds – but this needs spelling out better.

Thank you for these suggestions. We have made substantial changes to our discussion, streamlining it to focus on the most important parts and spending more time on the adaptive consequences of the ants' behavior. As discussed above, we would prefer to avoid imposing an explicit threshold structure in our model, and the data do not indicate clear thresholds for joining and leaving. Even without explicit thresholds, our accumulator model predicts that the system will result in hysteresis similar to that found in real bridges.

In conclusion, this study offers one of the few attempts to examine an interesting, charismatic, and potentially broadly applicable topic of dynamic response of superorganismic structures. I have no doubt that it would be read with interest by a broad range of researchers, and be of interest to the general public as well. However, I am not convinced that the main conclusions are supported by evidence, and have other misgivings as well (see minor comments below). I am thus not convinced that anything really new has been firmly demonstrated (hysteresis in collective behavior has been shown before), nor that any large insights have been gained. If the authors could assuage my doubts on these points, I would find this work to be a good fit for Nature Communications. If not, then not.

We thank Reviewer 2 for their helpful comments.

Minor comments

- Introduction: I was surprised that no mention was made of other collective behaviors in social insects, nor the frameworks used to understand them. Do the theoretical frameworks used to understand collective transport, or collective, dynamic choice of food sites, also offer a framework for this collective behavior?

Our initial manuscript did refer to weaver ant chains, fire ant rafts, and honey bee swarm structures in the first paragraph of the introduction, and we cited some of the theoretical works used to understand them. In our discussion we also referred to other social insect behavior theory, including response thresholds and integrator models (lines 322-325). Such theoretical frameworks inspired our work in the sense that they demonstrate how informative mechanistic models of behavior can be. However, these frameworks are not directly applicable to the control of army ant bridges, which in our estimation is fundamentally different from other, better-studied behaviors, and warranted the development of tailored mechanistic models. To keep the introduction relatively brief, we focused on discussing previous theoretical work on army ants, though we would be happy to add additional text on earlier theoretical work on other social insect behaviors if desired.

- Line 72 – ‘ground’ isn’t quite right. Substrate is better, but I admit, less catchy.

We have revised this paragraph in response to other comments, and this sentence has been removed.

- I didn’t find paragraph 79-91 very enlightening, as I really only understood them having read the manuscript and supplement.

We have revised this paragraph significantly, removing the list of questions and more succinctly describing that three sets of results will follow: experimental results for the structures as a whole, mechanistic modeling results, and experimental results at the individual level (lines 81-89). We have also changed the headings of results subsections to match (lines 91, 192, and 377).

- Ln 94 here, for traditionalists who like to know what was actually done before they read the interpretation, please specifically state that the detailed methods are found in S1.

We have added the following sentence: “Detailed methods are included in Methods: Field experiments” (line 94).

- Ln 128 – F – odd that droop is not maximum at the maximum extension, but rather at middle 15-20. This doesn’t seem to be explained anywhere, nor even mentioned.

During the expansion phase, the droop is indeed highest at the maximum extension. However, once the gap is then reduced, the bridge droops even more because ants do not immediately leave, and the structure is then longer than needed. This is one outcome of the hysteresis we describe. We have added an explanation of this in lines 161-165.

- Figure 3B – perhaps removing the extreme values in this figure (anything above 2 or 4) would make this a more useful figure? Similarly, the Y axis of C could be trimmed a bit to zoom in on the action.

We have made these changes as suggested (see new figures 3b and 3c).

- Lines 176-178 – nice, and interesting, but this is just a story unless there are data to back it up.

We have addressed this by adding new results showing that the hysteresis emerging in the accumulator model does indeed stabilize bridges experiencing small, momentary terrain shifts, as mentioned above (new Figs 4D-F and lines 283-298 and 342-354).

- Line 188 – no it isn't: there are clear dips at the start and starting at c. 20 seconds. Why is that? Is this a sampling artefact? See later comment as well.

The apparent dip at the start is simply due to the fact that there can be no events at negative times (0 is a hard boundary), and the apparent dip starting at c. 20 seconds is because each gap interval is only 30 seconds long. There are a few data points at longer time intervals, because we waited to expand a gap when the bridge was broken. We have clarified this in the caption to Figure S10.

- Line 231 – please explain briefly why.

We have added an explanation that this is due to asymmetry in the data-driven rates (line 256-260 and SI Figure S12).

- Line 290 – sorry, what was non-linear about this feedback?

Feedback in the accumulator model is nonlinear because the average number of joining or leaving events in each timestep (μ) depends nonlinearly on the dynamic deficit/excess variable ($\Delta(t)$) as shown in Figure 3C. We have clarified this in line 156 and 264.

- Line 337 – 339 or that a slack bridge affects your measurements somehow. Really, I don't see why flow was not measured directly - the traffic is not THAT high, and you hardly get any seriously overlapping ants. And even if you did, you could easily get the real count from either the top or the side. Inferring by proxy and software opens the door to lots of unexpected biases, and probably didn't save a lot of time in the end.

We agree that the level of overall traffic, *i.e.* the number of ants using the bridge, would be reasonable to measure directly. However, this would not be very informative about how functional the bridge is at any given time. To assess bridge performance we needed to look at variation in the flow rate of ants across the length of the platforms and bridge (*i.e.* across the x-dimension of the video frame). This allows us to compare the flow rate of ants on the bridge with the background flow rate of traffic on the trial. We did attempt to do this manually, but found it to be impractical. Our measure of performance is the residuals from regressions of bridge flow rate on platform flow rate (with a separate regression for each trial). We used this measure not only to account for overall traffic but also to minimize the potential effect that the varying background within the video frame (platform vs bridge) would have on the measurement.

- Figure 5(2) – Dot fill is not helpful. I can tell which side of the line each dot is on even without fill. Rather, you could use the fill for something more informative, like to see expansion vs contraction phase. ‘I’ is also missing in ‘brdge’ on Y axis

We have made these suggested changes (see Figure 5).

- Lines 384-5 – probably bridges with redundancy are more resilient to individual ants losing their grip. You could maybe test this: is a break more likely to happen on an expansion in sparser bridges?

The need for redundancy is a reasonable explanation for why ants would join bridges that are functioning well at all, but we are not sure it is a good explanation for why ants would be *more* likely to join high-functioning bridges than low-functioning bridges. In any case, we agree that it would be interesting to study the effects of bridge structural properties on the breaks. We did not do so as part of this project in part because this data set includes only 10 breaks, and in part because our focus was on the ants’ behavior under functional bridge conditions.

- Lines 431-439 – I am unconvinced that the tautness of the bridge is in any meaningful way affected by the ant traffic. Firstly, by simply examining the video we can see that the bridge doesn’t change at all when ants step on and off it. Secondly, the weight of a few ants compared to the strength of the bridge is so minimal that we should probably not expect to see an effect.

We did not mean to suggest that tautness is affected by ant traffic, rather our point is that *both* tautness *and* ant traffic should affect the tension experienced by ants in the bridge. We agree that the shape of the bridge doesn’t change when ants step on and off of it, and that we wouldn’t expect to see this. Yet we do expect that ants within the bridge can likely feel the added tension when they are supporting more traffic, just as one feels more tension in the arm when carrying a heavy object, without the shape of the arm being significantly affected. We have shortened and clarified this section of the manuscript (lines 480-486).

- Supplement – key information missing: how many colonies? How many trails? How long a pause between trials? Any reason to think that some chemical cues are left on the platform (eg a ‘build bridge here’ pheromone) which changes the behavior from first trial to second? What was the width of the bridge?

Thank you for pointing this out; we have added methodological details to the methods section (e.g. lines 568-574 and 583-595). We conducted the ten complete trials on five different days at five distinct locations—ranging from approximately 200 m to 1.7 km apart. Therefore up to five distinct colonies are represented. No more than three trials were conducted on any given day. Due to the fact that army ant colonies migrate through dense leaf litter and forest canopy most nights, we cannot be sure that we did not conduct trials with the same colony on multiple days, as we now explain in the methods.

After a given trial, we waited to begin the next until traffic returned to smooth flow, with no evidence of there having been a bridge. This occurred fairly quickly yet we waited at least several minutes. We saw no evidence that pheromones or other cues might cause changes in behavior from one trial to the next—in fact, ant behaviors and overall bridge dynamics were very similar in back-to-back trials. In particular, bridges did not begin to form until ants were unable to cross the gap (which began

occurring at about 5 mm) regardless of how many previous trials had been conducted that day. The width of the bridges varied from trial to trial, over time within a trial, and across the length of the bridges. We show an example of the top view of a bridge in Figure 1, which readers can use to get a sense of the width, and these data are also incorporated into our structural bridge metrics, particularly volume, which we estimated using the height and width at each x value.

- Sup – Cardstock should be mentioned in the field experiments bit

We have corrected this by adding it to the last paragraph of the field experiments section, which is now in the methods section of the main text (line 581).

- Sup – “For short periods of time (~30 seconds, depending...” a bit unclear: you went through the whole of all the videos like this, right?

We agree that the way we phrased this was confusing. Indeed, we went through the whole of all the videos like this. But in order to make the task more manageable we broke up the videos into chunks of approximately 30 seconds each, a detail that is perhaps unimportant. We have revised this statement to make it clearer (see line 611-617).

- Sup – “For each time step, we draw the number of ants that join and leave the bridge from normal distributions with the same mean and standard deviation as the data, as further explained for each model below.” Did the original data have a normal distribution? Why not draw it from the data directly? Or at least a model of the data?

We agree that it is more appropriate to draw from a distribution that is fitted to the data directly. We have revised all of our models to use fitted negative binomial distributions as described in lines 228, 332, 637-642, 669-674, and Figure 3. We chose to use negative binomial distributions because they are appropriate for count data, but do not have the stringent assumption of the Poisson distribution that the variance is equal to the mean. We have updated our model results and discussion to reflect this, though we note that this update did not produce any major changes in our results, and our conclusions are the same.

- Sup – “recovering from a break” how was this defined objectively?

We thank Reviewer 2 for pointing out that this was missing. We have added the following explanation (lines 695-701): “We determined that the bridge had recovered from a particular break by performing a blind, manual examination of the plot of bridge volume over time. Estimated volume rapidly spikes when a bridge is broken as ants that are unable to cross build up, and volume rapidly declines when recovering from a break as these extra ants cross and move on. We considered the bridge to have recovered from a break when the slope of the volume plot returned to a range more typical of functional bridges.”

- S10 – This data is a bit puzzling. Wouldn't we expect this to have a poisson-shaped distribution? Why does the number of events slip down towards the 30 second mark? Did you simply get lucky with 30 seconds

being more or less how long ants need to recover? Or is this something about preparing to move bridge (e.g. coming closer to it, putting hand on cog, etc)?

We would only expect a poisson-shaped distribution (or any distribution with a peak closer to zero) if the events were directly prompted by changes in gap size. The point of this figure is to show that this is not true, that the distribution is *not* inflated at low values, and instead events are roughly evenly distributed. Therefore, the movement itself does not seem to be a direct cue to join or leave the bridge. The ants do not need to recover at all from each individual change, but rather it is the accumulation of gap changes that requires adjustment, with factors associated with the structure (i.e. performance, perhaps tension, as described in the manuscript) rather than the perturbations themselves, acting as cues.

The number of events is lower both around 0 and around 30 because these are edge cases, and are subject to artifacts due to slight variation in the actual frequency of gap changes (which were performed manually in the field) as well as the fact that it takes a non-zero length of time to change the gap size. We have added an explanation in the caption of Figure S10 to clarify that the number of events *appears* to slip down closer to 25 seconds because of the smoothing. We confirmed that this early apparent drop in events is an artifact by plotting the following histograms, which show that regardless of the bin size, it is only the two most extreme bins, those including 0 and 30 seconds, which have substantially lower counts than the rest.

- S14 – what are the units of the Y axis? State explicitly what the circles are.

The Y axis of this figure shows trial ID, simply separating the data for different trials so that within-trial patterns are visible. We have clarified this in the caption for this figure (now SI Figure S18), and also in the caption for SI Figure S15, to which this also applies. We also now explicitly state in both captions that the circles show the mean flow immediately preceding actual joining and leaving events in each trial.

Reviewer #3 (Remarks to the Author):

The authors present a study of army ant bridges. I liked the experimental part of the work, where they lengthened and contracted gaps underneath the bridge, and showed that army ants exhibit hysteresis: they retain more ants during contraction than during expansion. These dynamic tests are very original, and reveal a new feature of ant rafts that is in common with many materials. I think the experimental work is publishable on its own and would make a good paper.

The experimental work is well described, but the theoretical exposition could be greatly improved. Despite reading quite a lengthy text, I am not convinced by the title of the work, stating that hysteresis is stabilizing for the rafts. I would have liked to see more details on how the various models were tested. The figures are too many and not clearly presented. The flow charts in the figures are difficult to read and I would have preferred equations. The parameters used in each model should be presented more clearly.

We have substantially improved the explanation and discussion of the theoretical work, including improving the description of parameters and providing more methodological details (Figure 3 and lines 209-232, 326-336, and 624-681). We have also reorganized and improved the figures, including replacing the flow charts with equations and pseudocode. These changes made the figures, and figure 3 in particular, much more readable, and we thank Reviewer 3 for the constructive comments.

We agree that we previously had only indirect evidence for hysteresis stabilizing the bridges. However, the title is now directly supported: we have extended our simulations to new conditions, finding that the hysteresis in the accumulator model maintains stable bridges during vibration-like gap shifts (back and forth), while the linear models produce bridges which quickly become unstable in these conditions (see 283-298 and 342-354).

line 70 What kind of things will disturb twigs and leaves while ants cross over them? Wind? Animals? What is the duration that a bridge needs to stay in position?

Indeed, wind and animal movements are good examples of disturbances. We agree that this kind of context was lacking, and we have added it (see lines 68-73). The duration that bridges need to be present is highly variable, as they remain for as long as traffic remains. Bridges have been continuously observed for up to 30 minutes both in this study and in earlier work (Reid et al. 2015), though there is reason to expect bridges to last for much longer. *E. hamatum* colonies frequently forage for full days (dawn to dusk, Teles Da Silva 1982), and with heavy traffic large portions of the trail are quite stable, it is reasonable to expect bridges to last up to the full duration of foraging in some cases. However, we did not find the maximum duration of bridges documented in the literature.

Teles Da Silva, M. Behaviour of army ants *Eciton burchelli* and *E. hamatum* (Hymenoptera, Formicidae) in the Belem region III. Raid activity. *Ins. Soc* **29**, 243–267 (1982).

102 The largest bridge in this study is composed of only 19 ants. It would be useful if this number were stated in the abstract to give the reader an idea of the scale involved.

19 was not the size of the largest bridge, but rather the average of the maximum number of ants in the 10 trials over the duration of the experiment. The largest functional bridge in fact had a maximum of 28 ants. We tried to add this to the abstract, but we were not able to fit it, because we also shortened the abstract dramatically to fit within the 150 word limit.

104 The authors state the dynamics are repeatable, but the figure shows high variability. Can they explain what the Pearson correlation says in this case?

Our intention was to assert that the overall patterns are similar across trials, not that there is little variation or that the quantitative details are the same. For example, in all of the trials ants began forming a bridge at a gap size of approximately 5 mm, the bridges grew in size and reached a relatively consistent maximum size (barring broken bridges), and then decreased in size. The hysteresis was also consistent across the trials for each of the measures we examined. We have clarified the text, stating specifically what we mean by saying the dynamics were similar (see lines 98-142).

Fig 1. The authors conduct experiments to a maximum gap of 30 mm. What are these units in ant body lengths? Why was the contraction and expansion rate conducted at the rates performed? Similarly what is the rationale for making the gap this length?

As requested, we have added more explanation of the rationale behind methodological choices (see lines 568-574). *E. hamatum* workers are highly polymorphic, ranging in size from a few millimeters to over a centimeter. Ant body length can be challenging to measure, and ant size is generally reported using other metrics, such as the width of the head. We were unable to find data on body length for *E. hamatum* in the literature. However, we were able to roughly estimate body length for 48 ants (12 of each size caste) from our videos. In this analysis body length ranged from 4 to 10 mm. We have therefore added to the manuscript that the 30 mm bridge is approximately 3 body lengths for major workers and more than 7 body lengths for some minor workers (line 573).

We changed the size of the gap at a rate of 1 mm every 30 seconds because this was gradual enough to avoid bridges frequently breaking, while fast enough to allow trials to be completed in a reasonable amount of time (line 568). We chose 30 mm as the maximum gap size because, as we now state in line 570, it is biologically-realistic length—earlier work found naturally occurring bridges spanning gaps up to 26 mm long (Garnier *et al.* 2013), and in another study bridges on artificial platforms grew to a maximum size of 30 mm (Reid *et al.* 2015). Our choice was also somewhat constrained by the length of trials—longer gaps require longer trials, and thus require steady traffic for longer periods.

Fig 2. The caption has a typo: "results metrics"

This figure suggests that only 10 trials were conducted and that at least 6 of the trials had breaks that might influence the data. Do the authors measurements of hysteresis still hold for the bridges did not break? Did

the breaks happen more in the extension or the contraction? More comment should be given here besides the caption.

Parts A and B are very difficult to read, especially the insets. A different color combination might be used.

We have changed “results metrics” to “result metrics” in the caption. As mentioned above, we also reduced the number of insets to make these figures easier to read. We prefer to leave the color combination as-is, to remain consistent with the color scheme for the expansion and contraction phases in all figures. We chose this color scheme because it is color-blind friendly. However, we lightened the shade of these colors in the backgrounds of panels A and B to further improve readability.

Indeed, trials in which there were no breaks still exhibited hysteresis, as one can see in Figure 2 (bottom inset for each panel) and SI Figures S2 - S8. Breaks occurred in both experimental phases—there were two trials with no breaks, 4 trials with breaks in the expansion phase only, 3 with breaks in the contraction only, and 1 with breaks in both phases. Regardless of the presence and timing of breaks, all trials had hysteresis in the same direction. We have added a discussion of this in lines 146-147 and 164-165.

165 This paragraph on hysteresis is very nice, but I think it should be in introduction. The authors might also note that there are few theoretical models that can explain hysteresis.

Thank you for this suggestion. We did move most of this paragraph to the introduction, but we found it quite difficult to incorporate smoothly. Since the introduction is before we present the result that we discovered hysteresis, it is confusing to have it there, in our view. We would prefer to leave this paragraph in its previous location, right after the presence of hysteresis in this system is shown, but we will reconsider at the editor’s discretion.

We are not sure what the reviewer means in saying that few models can explain hysteresis. We would be happy to add this to the manuscript, particularly if the reviewer can suggest a citation.

194 the authors should explain the data-driven rates of joining and leaving. Are these numbers reasonable? It seems like the numbers are 1-2 ants per minute.

Indeed, approximately 1 to 2 ants join and leave the bridges per minute, on average. More precisely, in each 30 second timestep of the expansion, the mean numbers of joiners and leavers are 1.13 and 0.67, respectively. In each 30 second timestep of the contraction, the mean numbers of joiners and leavers are 0.33 and 1.04, respectively. We have added this information to the manuscript (lines 229-232). We have also improved our descriptions of the models, as described above.

Fig 3. The authors should clearly derive and explain each of these models. The font is quite blurry and the equations are difficult to read. I am not clear how these models describe the dynamics on the bridge.

We have substantially revised Figure 3 to more carefully explain these models. Instead of flowcharts, we have included equations and pseudocode as suggested, and we have separated the model development

portion of this figure from the model results (which we moved to Figure 4). We think this greatly improves the readability of this figure. We have also added details of the models to the methods section (beginning on line 623).

Fig 3 units should be clarified. Is mean rate per minute in terms of number of ants?

We have clarified in the caption that this figure shows the distribution of the number of joining or leaving events per 30 sec gap interval.

200 I fundamentally do not understand the behavior for leaving and joining the model. I understand there is traffic of near constant flow rate. Do ants make decisions as they cross the bridge? Do all ants in the bridge make decisions at each time steps? How are these decisions made? Using a random variable?

Some of these questions are addressed in the Supplement, but I would have liked more numbers given in the text. It sounds like the rates of joining and leaving are taken from a normal distribution. The authors should still state what are the factors that lead to these numbers.

We expect that in real bridges ants decide to join as they are crossing or near a bridge, and that ants decide to leave when they are within a bridge. However, we do not explicitly model traffic or individual decisions (these are not individual-based models). Rather, we model the *number* of joining and leaving decisions that occur each timestep as a random variable, allowing us to simulate the number of ants in the bridge over time. We have moved the description of our models from the supplement to the methods section (beginning line 624), and have improved Figure 3 and our model discussion to be clearer. We thank the reviewer for their constructive comments on improving the clarity of this section.

The Table in the supplement is not labelled. It also has no units.

Because of our revisions to the models and descriptions we have deleted this table.

248 The nonlinear accumulator model should be renamed because it is simply not specific enough. There are an infinite number of accumulator models, and this one is proportional to the excess or deficit ants in the bridge. The title should reflect that.

Thank you for this suggestion. We believe that the more general “accumulator model” name has value as it relates this model to other nonlinear models and accumulator models used in the literature. We explored adding terms such as excess-deficit, but the name quickly becomes unwieldy. The linear models also have simple names (linear baseline, linear delay) so this keeps the model names consistent. For now we have left the name as is, but are open to suggestions from the editor.

There is much reliance on the supplement, which makes the paper difficult to read. The authors refer the reader to the burstiness in the Supplement.

We have moved most of the methods details from the supplement to a methods section in the main text. We agree that this makes the paper easier to read. We have removed the “burstiness” discussion and figure altogether. We concede that the supplement is still lengthy, especially in that it contains many figures. However, whereas before there were key methodological details in the supplement, we have now structured the supplement to be helpful to particularly interested readers but unnecessary to understand our main conclusions. We have not moved figures from the supplement to the main text because the main text already has quite a few figures, as Reviewer 3 noted above. We would be happy to do so, however, and will leave that to the editor’s discretion.

Figures 4-6 give three new variables: the extent of hysteresis, the performance metric, and the flow over bridge and panels. The units for the latter two are not given in the figure. These variables should also be clearly defined.

The extent of hysteresis is a unitless proportion, as we now explain in line 705 and the caption for SI Figure 11 (which was formerly Figure 4). The flow over bridges and platforms is measured as the optical flow in videos, which is in units of pixels per video frame. These units are not intuitive, and we are not sure it would be helpful to include this in the axes for figures. However, we have clarified this in the caption for Figure 5. This proxy of traffic is sufficient for our purposes because we are interested in traffic flow only as a relative metric (i.e., flow on bridges relative to platforms and flow at one time relative to another). Finally, the performance metric is also a measure of flow, though in this case it is the residuals from regressions of bridge flow over platform flow. Units for residuals are rarely provided in our experience, and we feel it would not be helpful to do so here. We agree that this should have been explained more clearly, however, and have included an explanation in the caption for Figure 6.

Overall, the paper has some interesting experimental results, but the theoretical work could have been better explained. It seems like there are some negative results here (linear models don't work). The accumulator model seems to have more promise, but it is qualitatively matching at best. The paper also suffers from length and over-reliance on the supplement, and many variables and measurements not clearly defined. Some of these are due to the constraints of the word limit for this journal. This paper may benefit from being split into several publishable pieces.

We thank Reviewer 3 for the constructive comments, which have improved the paper. While the linear models do not reproduce the system’s dynamics, it is important to include them to demonstrate that simpler mechanisms, which are also reasonable hypotheses, are not sufficient. We have also added theoretical experiments to better expose the ecological significance of hysteresis in stabilizing structures against smaller momentary changes. The experimental work and theoretical models together give a more indepth picture of the dynamic control that ant colonies are capable of, and we have substantially revised the text to make the supplement unnecessary for understanding the main conclusions.

Reviewer #4 (Remarks to the Author):

This paper investigates bridge formation in *Eciton hamatum* army ants. The authors present experimental and simulation results that demonstrate the presence of hysteresis in the structural properties of the bridges as the size of the gap they span is increased and decreased. Overall this is an interesting paper that I believe could be suitable for a general interest journal however I do think the authors need to improve the manuscript. In general, I found the manuscript to be quite long and not as focused on the key findings as I would expect for this journal. I believe the authors could improve the paper in this regard and it would find a broader audience. I have several other comments which can be found below.

We thank Reviewer 4 for their positive and helpful comments.

1. The motivation for the study in terms of the adaptive role of the observed collective behavior is fairly weak. The authors claim that coping with gaps that grow and shrink gradually "occurs frequently because they are built on twigs, leaves, and other unstable features". It's hard to imagine how the real challenges the ants face (in terms of unstable features) maps to the experimental set-up of a steadily increasing and reducing gap. I would like to see a much better justification that the observed hysteresis is biologically relevant or a case made that this investigation has relevance in other contexts, i.e. that the presence of structural memory is important even if the exact conditions simulated are not present in the natural environment.

We agree that our discussion of the adaptive and ecological context of this behavior was somewhat insufficient. We have improved this substantially, by providing more context both for the behavior itself and for our results, especially the adaptive value of hysteresis (e.g. lines 68-73, 283-298, 342-354 and 526-537). We thank Reviewer 4 and the other reviewers for pointing this out, as we feel our manuscript is really strengthened by addressing this.

Experiments which test study subjects with conditions outside of their typical experience are often needed to understand the underlying mechanisms of behavior. We agree it is likely unusual for ants to experience a gap steadily increasing and then steadily decreasing; nevertheless we were able to learn more from this experiment than we would have by simply observing bridges under typical conditions. By adding new simulations of our models under conditions of small, momentary shifts in gap size (vibration-like conditions), we now show in the revised manuscript just what the reviewer rightly pointed out was lacking. In this system, structural memory (hysteresis) is important because it allows bridges to respond to enduring shifts in the terrain while avoiding unnecessary adjustments to momentary shifts.

2. I don't believe the issue of whether the speed at which the gap is increased or decreased is properly addressed. The hysteresis could well be a transient phenomena that would disappear if the rate at which the gap changed was decreased. While I don't expect the authors to rerun experiments, I would like to see this investigated in the simulations. It wouldn't be too difficult to show that there are genuinely alternate stable states in the simulation model for a fixed gap size i.e. if the increase/decrease of the gap was stopped midway would an increasing trajectory reach a stable steady state that was different to a decreasing trajectory? I don't believe that whether the hysteresis is rate dependent or not affects the novelty of the study but it should be explored in more detail.

This is a helpful suggestion, and we have added a new sensitivity analysis of the behavior of our model to the rate (see lines 358-361 and Figure 4H). Hysteresis was still present in our simulations regardless of the rate, though the extent of hysteresis did vary, as the reviewer suggested. This is true for many systems with hysteresis; such systems are described as having rate-dependent hysteresis. When the rate of gap size change is increased (to 2 mm per timestep), the extent of hysteresis is larger, and when it is decreased (to 0.5 mm per timestep), the extent of hysteresis is smaller (Figure 4H repeated below). In principle, with an extremely small rate of gap-size change one would expect to see negligible hysteresis, though we should avoid drawing overly strong conclusions from these simulations as the data they are based on was collected at a single rate. While it may be rate-dependent, the hysteresis in this system is significant and adaptive, because it allows ants to respond only to those terrain shifts which require it, rather than responding to every momentary change. The rate of gap-size change used in our experiment exposed the hysteresis inherent in the real system, allowing us to develop more meaningful models that can account for the complex dynamic control that has evolved.

Minor comments

- I don't think the term 'dynamic control' would be well understood outside the engineering community. I would consider rewording the title and the abstract to make the paper more accessible.

Dynamic control is definitely a well known term in the engineering community, but it also has a general science connotation of adaptation to changing environments. Many social insect researchers (such as two of the authors) use this term in the context of collective behavior. We would prefer to leave this term in the title, as we think it is the best phrase to convey our meaning, though we are willing to reconsider at the editor's discretion.

L79 - I found this framing as a sequence of questions a bit odd - the 'how' of question 1 would seem to encompass questions 2 and 3, while the initial "Do the collective structures adapt.." seems redundant - we're already told they do on L68. I would much prefer if this was framed in terms of experimental results followed by simulation study.

We thank Reviewer 4 for this suggestion. As discussed above in the responses to Reviewer 2, we have substantially revised this paragraph, which now describes three sets of results: experimental results for the structures as a whole, mechanistic modeling results, and experimental results at the individual level (lines 81-89).

- I'm curious as to the role of bridges breaking in the process. In both panels in the figure there is a shaded region indicating a break and recovery. Presumably this only happens in the expansion phase, but does this always happen? Is there any relation between the occurrence of a break and the hysteresis volume?

In fact, breaks occurred in both the expansion and contraction phases. There were two trials with no breaks, 4 trials with breaks in the expansion phase only, 3 with breaks in the contraction only, and 1 with breaks in both phases. We agree that it would be interesting to rigorously study the role of breaks in bridge dynamics, but did not do so as part of this project, in part because our data contain a relatively small number of breaks (10), due to the fact that we necessarily focused on complete trials that maintained sufficient traffic for the duration of the experiment. Given the limitations of our data set, it would not be appropriate to attempt to draw conclusions about the effects of breaks.

- the figure insets are very small - I'm not sure they add much value.

We have reduced the number of figure insets to 3 and made them larger, in order to improve the readability of these figures.

L178 - the authors state "In army ant bridges, hysteresis may play a similar role, improving stability and preventing ants from constantly joining and leaving when the bridge is close to its optimal size." This is an interesting statement but it's not clear exactly how this would work. Could this be explained in more detail? Is this referring to a scenario when the gap is contracting or static? How would the hysteresis prevent ants from constantly joining/leaving, doesn't the hysteresis emerge from the decisions of the ants?

We have made substantive changes to our explanations and discussion of hysteresis and its implications throughout the manuscript, and the role and consequences of hysteresis in this system are now much clearer. These changes included revising the statement referenced above, it now says "...hysteresis may play a similar role, improving stability and allowing the structures to avoid unnecessary adjustments to small, momentary gap size changes." The original statement was referring to a static gap. For a static gap size of 20 mm, for example, there may initially be slightly more ants that join than needed. Some may then leave, leading to too few, then more may join. It is difficult to control systems in a way that avoids such oscillations around the optimal "setpoint". Control systems that avoid this kind of constant oscillation often do so through hysteresis.

I'm not convinced the inclusion of the linear models in the main text is necessary, especially as one model is included only for completeness (L202). The nonlinear model is the only one really of interest given the results so I would relegate the other models to the SI and include only to demonstrate that the nonlinearity is required.

We have revised our text to better explain why the linear models are relevant. We included the linear models because they represent reasonable first hypotheses for mechanisms of collective bridge formation. It is reasonable to expect the ants' response to be proportional to the perturbation (the gap-size change), as is the case for both linear models. It is also reasonable to expect that there would be a delay in ants leaving the bridge (delay model). While we previously stated that this model was included for "completeness", we see now that this wording was too vague. In fact, we included this model because one may intuitively expect it to result in hysteresis, which we now clarify (line 137). In fact, when we informally discussed our experimental results with colleagues, multiple people suggested that a delay could account for the hysteresis. It would be inappropriate not to consider these models before leaping to a more complex model without justification. We have kept these models in the main text, rather than the SI, and improved our text to demonstrate that these simpler first hypotheses are insufficient.

Furthermore, the linear models are actually more important in the revised version than in our original submission, because of the results of our new model extension, which tests the models' behavior under small, vibration-like shifts in terrain. Specifically, the contrast in the linear model results and the accumulator model results for these vibrations really illustrates why hysteresis stabilizes the bridges, in our view. We have revised our description of the models to make both of these points clearer (lines 209-212, 298-308, and 362-376).

L406 to L478 - this section is quite long and comes across as fairly speculative. Consider condensing to the a few key points.

We agree that this section was longer than necessary and have substantially reduced it (by about 40%) to focus on the main points (see lines 465-509).

L429 a reference is required here

After reducing the length of this section this sentence has been removed.

Peer review comments, second round review –

Reviewer #2 (Remarks to the Author):

McCreery et al. have done a good job in revising this manuscript, which is now substantially improved. The addition of the minor fluctuation model does indeed provide some support for the claim that hysteresis stabilises the bridges. The figures are easier to read, and the rationale for the work clearer, and the discussion clearer. I have no further technical concerns about this work.

I am still not convinced that we're getting any especially deep or broad insights into animal behaviour or the behaviour of collective systems, considering that hysteresis and its stable control benefits (line 538) have been demonstrated in many self-organised biological systems before (as the authors state). I would argue that being the "first to demonstrate decentralized, dynamic control and hysteresis in a self-assembled structure in social insects" is a pretty specific thing. However, this work can be seen as a proof of concept, that these army ant bridges are a useful model for examining such collective behaviours at an experimentally approachable scale. Perhaps in the future direct measurements on the tension acting on the ants (for example using calibrated springs as anchoring points) might offer more direct mechanistic support for the models.

Minor comments:

For additional clarity and to emphasise the key findings, a simple flow diagram for an individual ants' decisions might be helpful – what are ants checking in their attached and detached stages, and what are the effects of these check? This will probably be quite a simple diagram, but I think would really help. Only, of course, if space allows.

Line 477-478 – while a direct experimental test of the 'tension-affects-leaving' hypothesis must indeed wait, could support not be gained from the current data by, for example, looking at the leaving rates of ants expected to always be under the lowest tension (i.e. those at the bottom of the slack central part of the bridge) as compared to ones expected to be under high tension (ones close to the anchor point) or randomly selected ants?

Reviewer #3 (Remarks to the Author):

The authors have revised their paper on ant bridges. They have included the equations and graphs describing their model which help their case. Overall, I am more convinced of their main message that ants need to sense the excess or deficit of ants from equilibrium. I still have some suggestions that will help future readers below.

Fig 3 defines the number of ants joining and leaving the structure as n_j and n_l . Why are they described as negative binomial distributions? It seems that they could be fit by exponential distributions as well.

I can't understand the rationale for having 4 distinct distributions describing the expansion and contraction process. The gaps are being changed in size at a rate of 2 mm/min. Given this slow rate, I am surprised that ants can detect the detect the phase (expansion vs contraction). Thus I would expect there to be only two distributions (for leaving and joining, irregardless of phase). Lastly was there a control experiment performed with no gap motion?

Fig 3 should probably state the step size of 30 seconds so future workers can use this data.

Each distribution are characterized by mean μ and dispersion parameter θ . What are the confidence of fits for these distributions fitting the data? The fitting parameters should be given so that can be of use to other investigators. I see on line 231 the means are given but not the variance.

The nonlinear accumulator model doesn't depend on phase. The joining and leaving rates only

depend on Δt . However, Fig 3bc do not seem like very good fits. In 3b, the data points do not go through any of the curves. 3c also seems like a huge amount of scatter, without a clear trend as a function of Δt . In previous figure Fig 3a, the histograms were shown, and this figure also needs histograms to demonstrate how well the fit is. Lastly, the θ and μ values should be reported. It's not clear if these are the same values used as in Fig 3a.

It appears that there are much more extreme events of a deficit of ants rather than an excess. Why is that? Because of the hysteresis? Also the arrows for deficit/excess are incorrect.

Fig 3b nonlinear accumulator model states that the desired state of the bridge has ants that scale linearly with the gap size. This may be true for short bridges, but longer bridges, I would expect need to be stronger and thicker due to scaling of moment. At what size do ant bridges start to become thicker? It is larger than observed in this experiment?

I suggest splitting Fig 3 into two separate figures. That would help with clarity in the nonlinear accumulator model.

Fig S9 has no units in the axes labels.

Fig S10. I am a little confused at this distribution. The gap is changed every 30 seconds. So how is possible to obtain data past 30 second "time since gap change"?

line 251. the joining rate during expansion is 1.13 and the leaving rate during contraction is slightly less at 1.04. thus, it makes sense you won't get much hysteresis. the authors state that the joining and leaving rates were constant during each of the phases. thus, I don't understand why the linear model doesn't work. are one of the assumptions not true?

line 283 I like the simulation on the vibration, but the results seem to highly depend on the mean joining and leaving rates. Are the rates of 1.13 and 1.04 truly distinguishable? A t-test might show that these numbers are not significantly different.

Lastly, the title states that hysteresis stabilizes the bridge, but the authors have merely shown that an excess-deficit model is a good fit for their data. There may be many more models that lead to good fits of the data, including those that do not result in hysteresis. How can the authors be so sure that hysteresis is the main characteristic of their model that leads a good fit? In short, I was convinced that their model describes their data well, but not convinced that all such models that fit the data will exhibit hysteresis.

Reviewer #4 (Remarks to the Author):

The authors have adequately addressed my concerns and I'm happy for this paper to be accepted

For the following, reviewers' comments are in black, and our responses are in blue.

Reviewer #2 (Remarks to the Author):

McCreery et al. have done a good job in revising this manuscript, which is now substantially improved. The addition of the minor fluctuation model does indeed provide some support for the claim that hysteresis stabilises the bridges. The figures are easier to read, and the rationale for the work clearer, and the discussion clearer. I have no further technical concerns about this work.

We thank Reviewer 2 for their thoughtful consideration and comments.

I am still not convinced that we're getting any especially deep or broad insights into animal behaviour or the behaviour of collective systems, considering that hysteresis and its stable control benefits (line 538) have been demonstrated in many self-organised biological systems before (as the authors state). I would argue that being the "first to demonstrate decentralized, dynamic control and hysteresis in a self-assembled structure in social insects" is a pretty specific thing. However, this work can be seen as a proof of concept, that these army ant bridges are a useful model for examining such collective behaviours at an experimentally approachable scale. Perhaps in the future direct measurements on the tension acting on the ants (for example using calibrated springs as anchoring points) might offer more direct mechanistic support for the models.

Minor comments:

For additional clarity and to emphasise the key findings, a simple flow diagram for an individual ants' decisions might be helpful – what are ants checking in their attached and detached stages, and what are the effects of these check? This will probably be quite a simple diagram, but I think would really help. Only, of course, if space allows.

We thank Reviewer 2 for this suggestion. We have added this flow diagram as Figure 7 (line 495).

Line 477-478 – while a direct experimental test of the 'tension-affects-leaving' hypothesis must indeed wait, could support not be gained from the current data by, for example, looking at the leaving rates of ants expected to always be under the lowest tension (i.e. those at the bottom of the slack central part of the bridge) as compared to ones expected to be under high tension (ones close to the anchor point) or randomly selected ants?

We agree with Reviewer 2 that an analysis of the locations of leaving ants relative to those that stay may provide indirect support for this hypothesis. However, while we have precise data for the timing of leaving events, we do not have precise data for the locations within the bridge of those leaving events. Location data is difficult to record because it is hard to distinguish among ants within the 3 dimensional bridge from just the two camera angles. We did record approximate location data for portions of the experiments, and these data suggest that leaving ants are more likely to come from the middle of the bridge than the edges, consistent with the hypothesis. However, this could simply be because there are likely more ants total in the middle of bridges than the edges. To rigorously evaluate whether leaving ants were more centrally located than expected by chance, one would ideally need to know the total

number of ants in each part of the bridge at all times—data which we do not have. We agree that a custom designed experimental setup to directly and rigorously test the tension hypothesis is an interesting future direction

Reviewer #3 (Remarks to the Author):

The authors have revised their paper on ant bridges. They have included the equations and graphs describing their model which help their case. Overall, I am more convinced of their main message that ants need to sense the excess or deficit of ants from equilibrium. I still have some suggestions that will help future readers below.

Fig 3 defines the number of ants joining and leaving the structure as n_j and n_l . Why are they described as negative binomial distributions? It seems that they could be fit by exponential distributions as well.

We used the negative binomial distribution because it is a discrete probability distribution, appropriate for count data, and n_j and n_l are counts. Our count data cannot be represented by the continuous exponential distribution. We also note that the negative binomial distribution is actually a special case of the exponential distribution for count data.

I can't understand the rationale for having 4 distinct distributions describing the expansion and contraction process. The gaps are being changed in size at a rate of 2 mm/min. Given this slow rate, I am surprised that ants can detect the detect the phase (expansion vs contraction). Thus I would expect there to be only two distributions (for leaving and joining, irregardless of phase). Lastly was there a control experiment performed with no gap motion?

We do not mean to suggest that the ants are necessarily detecting the difference between the expansion and contraction phases, however we also do not want to make a restrictive assumption that the two phases behave exactly the same. We fit separate distributions for the expansion and contraction phases partly to avoid making such an assumption. To implement the linear models, we needed to simulate the number of each type of event in each phase. Therefore we estimated from the data what the distributions we draw from should look like, letting the data speak for itself. However similar the expansion and contraction distributions are in the real ants, our estimated distributions will reflect that similarity. While we use four distributions to parameterize our models, we do not state that one needs four distributions in order to fit the data (and the results of these linear models do not fit the data well anyway). We simply chose to allow separate joining and leaving in both phases to *allow the possibility* that ants behave differently when they experience expanding gaps (overstretched bridges) vs decreasing gaps (slack bridges). Indeed, Reviewer 3's intuition is correct—in the best performing model (nonlinear accumulator model), the ants do not need to distinguish phase because the model draws directly from the data for each deficit value.

Finally, we did not perform a control, per se, in which we evaluated static bridges. All of the previous studies of army ant bridges have focused on bridges with static terrain geometries. Our goal was rather to look at dynamics under changing terrain geometry, and to focus on response to perturbations. Because we are not asking whether our manipulation (changing the gap size) has an effect on the bridges at all, but rather we are characterizing the nature of the systems' response, we did not use static bridges as a control.

Fig 3 should probably state the step size of 30 seconds so future workers can use this data.

We have added this to figure 3 and we thank Reviewer 3 for this suggestion. The model descriptions now begin with “For each 30 second timestep, t , ...”

Each distribution are characterized by mean μ and dispersion parameter θ . What are the confidence of fits for these distributions fitting the data? The fitting parameters should be given so that can be of use to other investigators. I see on line 231 the means are given but not the variance.

We thank Reviewer 3 for pointing out that the values of the dispersion parameters were missing. We have added these θ values to the manuscript in lines 231-233. Regarding the confidence of the fits, Figure 3a shows both the raw data (filled bars) as well as the fitted distributions (bold outlines). In each of the four cases, there is nearly perfect overlap between the data and fitted distribution (overlapping portion shown as paler fill).

The nonlinear accumulator model doesn't depend on phase. The joining and leaving rates only depend on Δt . However, Fig 3bc do not seem like very good fits. In 3b, the data points do not go through any of the curves. 3c also seems like a huge amount of scatter, without a clear trend as a function of Δt . In previous figure Fig 3a, the histograms were shown, and this figure also needs histograms to demonstrate how well the fit is. Lastly, the θ and μ values should be reported. Its not clear if these are same values used as fig 3a.

Figure 3b shows binary data, zeros and ones (with some vertical jitter for clarity), while the curves show the results of a logistic regression on those data, i.e. the probability of success (of a “one”, or a joining/leaving event) at different packing densities. The curves are not meant to go through the data, but rather reflect the relative number of zeros and ones across the x axis. In this case, the curve showing the probability of leaving is higher at high packing density because there were more leaving events at high packing densities than low, and the opposite is true for the joining probability. This type of figure is the most common way of showing the results of a logistic regression, to our knowledge (described in Whitlock 2015, and elsewhere). We state in the caption what the curves and dots represent, and we hope that this will mitigate potential confusion among readers.

Regarding Figure 3c, we agree that the number of events per gap interval is a noisy metric, and thus there is a large amount of variation even within a particular deficit value. We do not intend to say that our accumulator model captures all of the variation in number of joining and leaving events, we just use this analysis to parameterize our model. Indeed, despite this scatter, the accumulator model performs well, reproducing the most salient aspects of collective bridge dynamics. Regarding reporting the values of θ and μ for this figure, these are not the same values as shown in Figure 3a. For Figure 3c, μ is a function of the deficit, we report the relationship in the methods section (lines 692-695), where we also report the θ values. We have now also added these details to the results section (lines 337-339).

Whitlock, M., & Schluter, D. (2015). *The analysis of biological data* (p. 768). Roberts Publishers.

It appears that there are much more extreme events of a deficit of ants rather than an excess. Why is that? Because of the hysteresis? Also the arrows for deficit/excess are incorrect.

We assume that Reviewer 3 is referring to the arrows below Figure 3c, which show the direction, along the x axis, of increasing deficit vs increasing excess in ants. These arrows are, in fact, correct. The x axis of this figure shows the deficit, which is positive when the bridge has fewer ants than the equilibrium, and negative when there are excess ants. We acknowledge that this is a bit counterintuitive. We intended the arrows to clarify this, but see that it was still somewhat confusing. Therefore, we have also changed the axis title to “deficit in number of ants”, instead of “deficit/excess” to make it unambiguous.

Therefore, rather than there being much more extreme deficit events in our data, there are instead more extreme instances of *excess* ants. Specifically, while it was somewhat rare in our experiments for a bridge to have a deficit of more than 5 ants, it was not uncommon for a bridge to have an excess of about 10 or more ants. This is likely because if the deficit grows too large a bridge will fail. On the other hand, while a very large excess (i.e. highly negative deficit) may be inefficient, it is unlikely to lead to bridge failure.

Fig 3b nonlinear accumulator model states that the desired state of the bridge has ants that scale linearly with the gap size. This may be true for short bridges, but longer bridges, I would expect need to be stronger and thicker due to scaling of moment. At what size do ant bridges start to become thicker? It is larger than observed in this experiment?

This is a good question, and in fact the bridges in our experiments were somewhat thicker at their maximum length than when they were substantially shorter. The bridge thickness (or mean cross-sectional area) is even a little more complex, as we found it also has hysteresis and depends on the phase. However, we cannot tell whether these differences reflect different “desired” states, or simply different realized states from experimental dynamics. In other words, we don’t have enough information to know exactly what the scaling relationship is, and we have to make some simplification in order to parameterize the model.

We chose the simplest option of linear scaling, as Reviewer 3 pointed out. It’s possible that our model would fit the data better if we were to use a more complicated scaling relationship, but this would require making assumptions about the precise nature of this relationship. We don’t feel we have enough data to justify such assumptions. Using linear scaling is a simple, first approximation, which as it turns out, still reproduces the salient behaviors of the ants quite well.

I suggest splitting Fig 3 into two separate figures. That would help with clarity in the nonlinear accumulator model.

We thank the reviewer for this suggestion. Figure 3 is already separated such that the nonlinear model has its own panel, and we think it may be helpful to readers to be able to compare these models side-by-side. Therefore, we have left it as one figure in this submission, but would be happy to reconsider at the editor’s discretion.

Fig S9 has no units in the axes labels.

Figure S9 shows traffic flow. As with other figures showing optic flow (discussed in responses to reviewer 3's comment about Figures 4-6 in the previous round of review), the units here are fairly non intuitive, and we are not sure it would be helpful to include in the figure axes. The traffic flow over bridges and platforms is measured using the proxy of optical flow, which is in units of pixels per video frame. This proxy of traffic is sufficient for our purposes because we are interested in traffic flow only as a relative metric. We have clarified this in the caption for Figure S9. Specifically, the caption now includes that "optic flow measures the amount of movement from one video frame to the next, in units of pixels per frame."

Fig S10. I am a little confused at this distribution. The gap is changed every 30 seconds. So how is possible to obtain data past 30 second "time since gap change"?

While the gap was nearly always changed every 30 seconds, some intervals lasted longer if the bridge was non-functional (as stated in the methods and the caption for this figure). There were, therefore, a small number of events which occurred more than 30 seconds after a gap change.

line 251. the joining rate during expansion is 1.13 and the leaving rate during contraction is slightly less at 1.04. thus, it makes sense you won't get much hysteresis. the authors state that the joining and leaving rates were constant during each of the phases. thus, I don't understand why the linear model doesn't work. are one of the assumptions not true?

We agree that the results of the linear models' simulations are not unexpected, given the similarities between these expected numbers of events. And indeed, the reviewer is correct that the linear models fail because one of their assumptions is not true: specifically, the assumption of a linear response. This is precisely why we need the nonlinear accumulator model. We state that the rates of joining and leaving were constant during each phase *in the linear models*. The models fit poorly because these rates of joining and leaving were not constant in the experiments with real ants. However, we see now that this may have been confusing, because we also state that the rate of perturbation was constant throughout the real experiments at 1 mm per 30 seconds. We have clarified that the constant probabilities of joining and leaving is a reasonable *assumption* of the linear models, rather than something known about the system (line 210).

line 283 I like the simulation on the vibration, but the results seem to highly depend on the mean joining and leaving rates. Are the rates of 1.13 and 1.04 truly distinguishable? A t-test might show that these numbers are not significantly different.

Thank you for this comment. We see that the way we explained the asymmetry in joining and leaving could have been clearer. Rather than directly comparing 1.13 and 1.04 (the joining expected numbers of joining events in the expansion and leaving events in the contraction, respectively), the more salient comparison is the *net* number of ants the bridge gains during the expansion ($1.13 - 0.67 = 0.46$) vs the *net* number of ants lost during the contraction ($1.04 - 0.33 = 0.71$). We have clarified this in lines 260 and 280.

Reviewer #3's larger point still stands, however, and we agree that this slight difference may not represent a true, biologically significant asymmetry. Nevertheless, the fact that even a small asymmetry in the linear models makes highly unstable bridges demonstrates that these models fail to adequately capture the ants' behavior. We describe this in lines 306-309 and have further clarified this point in line 297-298. If the ants' self-assembly relied on perfect symmetry, it would not be a robust system. Furthermore, the vibrations result in extreme variability in linear simulations of bridges, even without asymmetry. On the other hand, the nonlinear accumulator model produces bridges that are robust to all of these potential issues, as we discuss in the paragraph beginning on line 368.

Lastly, the title states that hysteresis stabilizes the bridge, but the authors have merely shown that an excess-deficit model is a good fit for their data. There may be many more models that lead to good fits of the data, including those that do not result in hysteresis. How can the authors be so sure that hysteresis is the main characteristic of their model that leads a good fit? In short, I was convinced that their model describes their data well, but not convinced that all such models that fit the data will exhibit hysteresis.

In addition to showing that an excess-deficit model fits our data well, we also show that such a model leads to robust, high-performance structures. The excess-deficit model also makes sense in the context of this system, in that it uses information one can reasonably expect ants to be able to sense, does not require precise information about the terrain, and is simple to implement.

Nevertheless, we agree that there are other models that may fit the data, and we do not assert that the ant self-assembly uses exactly this model. Where we respectfully disagree with Reviewer 3 is with the idea that there are other models that will fit the data well *that do not result in hysteresis*. Our experiments consistently showed that hysteresis is inherent in the real system. A model that does not produce hysteresis will necessarily poorly fit the hysteretic data. It's not so much that hysteresis in our model leads to a good fit, but rather, hysteresis is a defining feature of the dynamics of the ants' self-assembly. We are able to reproduce this hysteresis with our model, and we show that it has the benefit of making the system robust and stable. We have added text to our conclusions section to clarify that hysteresis is an important feature of the real system, which we were able to reproduce (line 547-548).

Reviewer #4 (Remarks to the Author):

The authors have adequately addressed my concerns and I'm happy for this paper to be accepted

We thank Reviewer 4 for their positive remarks and helpful comments on the earlier version.

Peer review comments, third round review –

Reviewer #3 (Remarks to the Author):

The authors have revised their manuscript on ant bridges. The responses are helpful, but I still have a few points in the manuscript that should be made more clear.

Fig 3. I'm glad that the authors now use include the 30 second gap interval, but I would encourage them to use similar notation throughout the paper. Sometimes they call it "step size" and other times "gap interval"

Fig 3b. Thanks for the nice explanation on logistic regression, which is the first time I have heard of such a regression. Especially since this paper is going to a mixed audience, I suggest referencing the Whitlock reference the first time introducing the logistic regression.

Fig 3C. I suggest labelling the x-axis with simply "deficit of ants." I would also add an explanatory note in text or caption that a negative deficit corresponds to an excess. However, the notation $\Delta(t) = \text{deficit/excess}$ is simply confusing. I suggest also deleting the double arrows under the graph to reduce confusion.

The authors provided a good explanation in the response letter about the why a greater magnitude of excess ants is observed than deficit ants. I suggest that go in the text as well.

line 231 -- the authors should use the mathematical symbol for theta here

I like the flow chart included in Figure 7. It would be useful if the authors could include some kind of schematic of ants making these decisions and labelling leavers and joiners. That would help readers visualize the three models.

453 density is misspelled

Fig 5. the y-axis should have units. In the supplement, the authors wrote "optic flow" in many places which helped. There are two platform segments that are listed. Are they both used for measuring the flow rate?

In what scenarios would there be less flow over the bridge than the platforms and vice versa? Saying that explicitly would help to understand why this is a measure of performance.

Fig 5. What is plantform vs platform? This is an example of the many typos in this document that should be corrected.

Fig 6. there is no "dark-gray shading" as discussed in the caption. Also, what are the units of the y-axis in this figure? I still can't understand this figure or the accompanying text although I have read this paper several times. Why does this graph show that joining events are preceded by prior high performance?

For the following, reviewers' comments are in black, and our responses are in blue.

Reviewer #3 (Remarks to the Author):

The authors have revised their manuscript on ant bridges. The responses are helpful, but I still have a few points in the manuscript that should be made more clear.

Fig 3. I'm glad that the authors now use include the 30 second gap interval, but I would encourage them to use similar notation throughout the paper. Sometimes they call it "step size" and other times "gap interval"

We could not find an instance of the phrase "step size" in our manuscript, though we do sometimes use the term "timestep". We assume the reviewer is referring to this. We typically use "timestep" when referring to the model (in which there are discrete timesteps, and we do not model behavior between timesteps), and "gap interval" when referring to the real experiments (in which there were discrete actions occurring at regular intervals, but the experiments still happen in continuous time, and we have data continuously throughout a gap interval). Still, we see that this terminology was not clear enough in all cases, and we thank Reviewer 3 for pointing this out. To clarify this, we have added to the manuscript that "Each simulation timestep corresponds to a 30-second gap interval in the real experiments" (line 177).

Fig 3b. Thanks for the nice explanation on logistic regression, which is the first time I have heard of such a regression. Especially since this paper is going to a mixed audience, I suggest referencing the Whitlock reference the first time introducing the logistic regression.

We have added this reference as suggested (line 577).

Fig 3C. I suggest labelling the x-axis with simply "deficit of ants." I would also add an explanatory note in text or caption that a negative deficit corresponds to an excess. However, the notation $\Delta(t) = \text{deficit/excess}$ is simply confusing. I suggest also deleting the double arrows under the graph to reduce confusion.

Thank you for the suggestion, we have made these changes (see figure 3).

The authors provided a good explanation in the response letter about the why a greater magnitude of excess ants is observed than deficit ants. I suggest that go in the text as well.

We have added this to the manuscript in lines 272-276.

line 231 -- the authors should use the mathematical symbol for theta here

We have made this correction (line 181).

I like the flow chart included in Figure 7. It would be useful if the authors could include some kind of schematic of ants making these decisions and labelling leavers and joiners. That would help readers visualize the three models.

We thank Reviewer 3 for this suggestion. We agree it could be helpful to have a diagram of a bridge of ants, labeling potential joiners and leavers. However, getting the high-quality, professional graphics for such a diagram, has proven to be very difficult at this point in the process. Commissioning artwork would be quite time consuming. As a possible alternative, we attempted to find a screenshot from one of our videos of experiments to label, but the image quality of screenshots of videos is always a bit blurry, and we weren't able to find one that we were happy with. We have left Figure 7 as is, without the schematic, as we think it is clearer in its current form than a lower-quality schematic would be, but we are happy to reconsider at the editor's request.

453 density is misspelled

We're not sure which reference to density Reviewer 3 is referring to here, we haven't been able to find a case where it's misspelled. On line 452, we did use the word "densities" (now line 367), but this is intentionally plural. We are, of course, more than happy to fix any typos, and we thank Reviewer 3 for their careful read of the manuscript.

Fig 5. the y-axis should have units. In the supplement, the authors wrote "optic flow" in many places which helped. There are two platform segments that are listed. Are they both used for measuring the flow rate?

As described in previous responses, we do not think that the units of optic flow are intuitive or particularly helpful. We have, however, added "optic flow" so the y axis label now reads "optic flow over bridge" and the x axis label reads "optic flow over platform segments". Both platform segments were used for measuring performance, we have clarified this in line 340.

In what scenarios would there be less flow over the bridge than the platforms and vice versa? Saying that explicitly would help to understand why this is a measure of performance.

Thank you for this suggestion. We have addressed this by adding the following to the manuscript (lines 342-344): "If a bridge slows down the flow of ants to a greater extent than usual, the performance measure will be low; high performance reflects bridges which do not slow down traffic or do so less than usual."

Fig 5. What is plantform vs platform? This is an example of the many typos in this document that should be corrected.

We thank Reviewer 3 for pointing out this typo, which we have corrected (see figure 1, text "1. Flow rate of ants (optic flow) extracted separately for bridge region and *platform* regions"). We have carefully proofread the manuscript to correct any remaining typos.

Fig 6. there is no "dark-gray shading" as discussed in the caption. Also, what are the units of the y-axis in this figure? I still can't understand this figure or the accompanying text although I have read this paper several times. Why does this graph show that joining events are preceded by prior high performance?

The dark gray shading refers to the small areas underneath the distribution curves at the very tails (example circled in purple below). The y axis is unitless, because these are density plots (similar to histograms). We have clarified this in the caption by saying "Distributions (density plots) show the mean..."

The distributions in this figure show the relative probability of a random sample of timepoints from our dataset having the mean performance shown on the x axis. In other words, it shows the relative probability of different mean performance values, if the timing of joining and leaving events were *random*. For example, for joining events during the expansion (the upper, orange curve in the left panel), we can see that it's relatively likely that *random* events would be preceded by a mean performance of 2, because this is close to the peak of the distribution. It is very unlikely that random events would be preceded by a mean performance of -1.5, because this is way out on the tail of the distribution. Because the *actual* mean performance preceding joining events in our data is also way out on the tail of the distribution (this is the circle with crosshairs), we know that these events are *not* randomly distributed with respect to performance—that ants are significantly more likely to join bridges with higher performance. This is described in the manuscript in the Methods section (lines 633-644) and in the caption to Figure 6 (lines 881-887). We have also further clarified this analysis, as well as the location of the dark gray shading and the y axis, in the figure caption.